# CLIP-Dissect: Automatic Description of Neuron Representations in Deep Vision Networks

**Tuomas Oikarinen**
UCSD CSE
toikarinen@ucsd.edu

**Tsui-Wei Weng**
UCSD HDSI
lweng@ucsd.edu

## Abstract

In this paper, we propose CLIP-Dissect, a new technique to automatically describe the function of individual hidden neurons inside vision networks. CLIP-Dissect leverages recent advances in multimodal vision/language models to label internal neurons with open-ended concepts without the need for any labeled data or human examples. We show that CLIP-Dissect provides more accurate descriptions than existing methods for last layer neurons where the ground-truth is available as well as qualitatively good descriptions for hidden layer neurons. In addition, our method is very flexible: it is model agnostic, can easily handle new concepts and can be extended to take advantage of better multimodal models in the future. Finally CLIP-Dissect is computationally efficient and can label all neurons from five layers of ResNet-50 in just 4 minutes, which is more than $10\times$ faster than existing methods. Our code is available at https://github.com/Trustworthy-ML-Lab/CLIP-dissect.

## 1 Introduction

Deep neural networks (DNNs) have demonstrated unprecedented performance in various machine learning tasks spanning computer vision, natural language processing and application domains such as healthcare and autonomous driving. However, due to their complex structure, it has been challenging to understand why and how DNNs achieve such great success across numerous tasks. Understanding how the trained DNNs operate is essential to trust their deployment in safety-critical tasks and can help reveal important failure cases or biases of a given model.

One way towards understanding DNNs is to inspect the functionality of individual neurons, which is the focus of our work. This includes methods based on manual inspection (Erhan et al., 2009; Zeiler & Fergus, 2014; Zhou et al., 2015; Olah et al., 2017; 2020; Goh et al., 2021), which provide high quality explanations and understanding of the network but require large amounts of manual effort. To address this issue, researchers have developed automated methods to describe the functionality of individual neurons, such as Network Dissection (Bau et al., 2017) and Compositional Explanations (Mu & Andreas, 2020). In (Bau et al., 2017), the authors first created a new dataset named *Broden* with pixel labels associated with a pre-determined set of concepts, and then use *Broden* to find neurons whose activation pattern matches with that of a pre-defined concept. In (Mu & Andreas, 2020), the authors further extend Network Dissection to detect more complex concepts that are logical compositions of the concepts in *Broden*. Although these methods based on Network Dissection can provide accurate labels in some cases, they have a few major limitations: (1) They require a densely annotated dataset, which is expensive and requires significant amount of human labor to collect; (2) They can only detect concepts from the fixed concept set which may not cover the important concepts for some networks, and it is difficult to expand this concept set as each concept requires corresponding pixel-wise labeled data.

To address the above limitations, we propose CLIP-Dissect, a novel method to automatically dissect DNNs with *unrestricted* concepts *without* the need of any concept labeled data. Our method is training-free and leverages pre-trained multi-modal models such as CLIP (Radford et al., 2021) to efficiently identify the functionality of individual neuron units. We show that CLIP-Dissect (i) provides high quality descriptions for internal neurons, (ii) is more accurate at labeling final layer neurons where we know the ground truth, and (iii) is $10\times$-$200\times$ more computationally efficient

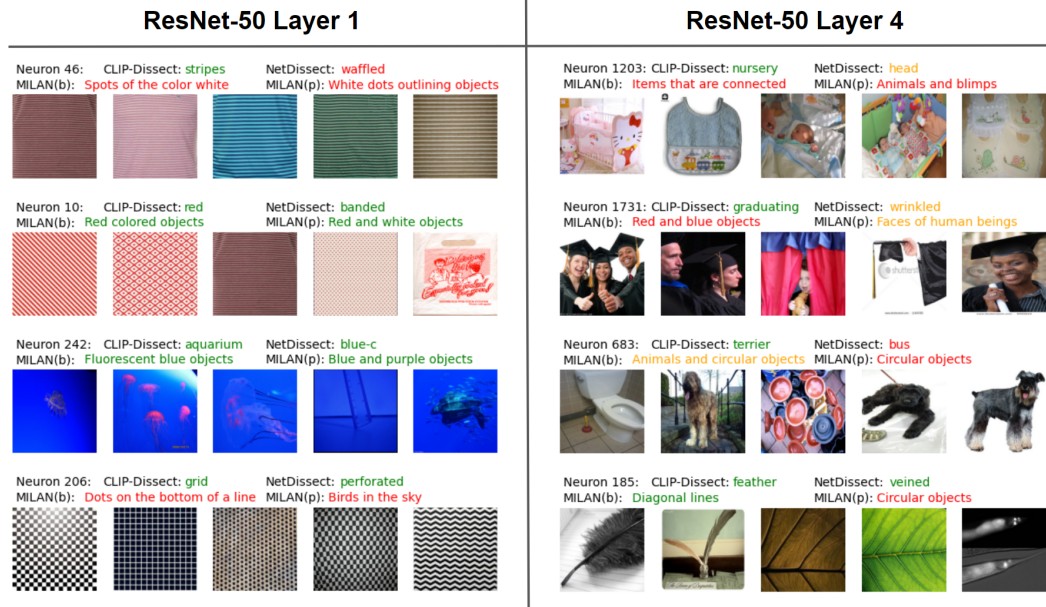

Figure 1: Labels generated by our method CLIP-Dissect, NetDissect (Bau et al., 2017) and MI-LAN (Hernandez et al., 2022) for random neurons of ResNet-50 trained on ImageNet. Displayed together with 5 most highly activating images for that neuron. We have subjectively colored the descriptions green if they match these 5 images, yellow if they match but are too generic and red if they do not match. In this paper we follow the *torchvision* (Marcel & Rodriguez, 2010) naming scheme of ResNet: Layer 4 is the second to last layer and Layer 1 is the end of first residual block. MILAN(b) is trained on both ImageNet and Places365 networks, while MILAN(p) is only trained on Places365.

than existing methods. Finally, we show how one can use CLIP-Dissect to better understand neural networks and discover that neurons connected by a high weight usually represent similar concepts.

## 2 BACKGROUND AND RELATED WORK

**Network dissection.** Network dissection (Bau et al., 2017) is the first work on understanding DNNs by automatically inspecting the functionality (described as *concepts*) of each individual neuron[1]. They identify concepts of intermediate neurons by matching the pattern of neuron activations to the patterns of pre-defined concept label masks. In order to define the ground-truth concept label mask, the authors build an auxiliary densely-labeled dataset named *Broden*, which is denoted as $\mathcal{D}_{\text{Broden}}$. The dataset contains a variety of pre-determined concepts $c$ and images $x_i$ with their associated pixel-level labels. Each pixel of images $x_i$ is labeled with a set of relevant concept $c$, which provides a ground-truth binary mask $L_c(x_i)$ for a specific concept $c$. Based on the ground-truth concept mask $L_c(x_i)$, the authors propose to compute the intersection over union score (IoU) between $L_c(x_i)$ and the binarized mask $M_k(x_i)$ from the activations of the concerned $k$-th neuron unit over all the images $x_i$ in $\mathcal{D}_{\text{Broden}}$: $\text{IoU}_{k,c} = \frac{\sum_{x_i \in \mathcal{D}_{\text{Broden}}} M_k(x_i) \cap L_c(x_i)}{\sum_{x_i \in \mathcal{D}_{\text{Broden}}} M_k(x_i) \cup L_c(x_i)}$.

If $\text{IoU}_{k,c} > \eta$, then the neuron $k$ is identified to be detecting concept $c$. In (Bau et al., 2017), the authors set the threshold $\eta$ to be 0.04. Note that the binary masks $M_k(x_i)$ are computed via thresholding the spatially scaled activation $S_k(x_i) > \xi$, where $\xi$ is the top 0.5% largest activations for the neuron $k$ over all images $x_i \in \mathcal{D}_{\text{Broden}}$ and $S_k(x_i)$ has the same resolution as the pre-defined concept masks by interpolating the original neuron activations $A_k(x_i)$.

(Bau et al., 2020) propose another version of Network Dissection, which replaces the human annotated labels with the outputs of a segmentation model. This gets rid of the need for dense annotations

---

[1] We follow previous work and use "neuron" to describe a channel in CNNs.

in $D_{probe}$, but still requires dense labels for training the segmentation model and the concept set is restricted to the concepts the segmentation model was trained on. For simplicity, we focus on the original Network Dissection algorithm (with human labels) in this work unless otherwise mentioned.

**MILAN.** MILAN (Hernandez et al., 2022) is a contemporary automated neuron labeling method addressing the issue of being restricted to detect predefined concepts. They can generate unrestricted descriptions of neuron function by training a generative images-to-text model. The approach of (Hernandez et al., 2022) is technically very different from ours as they frame the problem as learning to caption the set of most highly activating images for a given neuron. Their method works by collecting a dataset of human annotations for the set of highly activating images of a neuron, and then training a generative model to *predict* these human captions. Thus, MILAN requires collecting this curated labeled data set, which limits its capabilities when applied to machine learning tasks outside this dataset. In contrast our method does not require any labeled data for neuron concepts and is *training-free*.

**CLIP.** CLIP stands for Contrastive Language-Image Pre-training (Radford et al., 2021), an efficient method of learning deep visual representations from natural language supervision. CLIP is designed to address the limitation of static softmax classifiers with a new mechanism to handle *dynamic* output classes. The core idea of CLIP is to enable learning from practically unlimited amounts of raw text, image pairs by training an image feature extractor (encoder) $E_I$ and a text encoder $E_T$ simultaneously. Given a batch of $N$ image $x_i$ and text $t_i$ training example pairs denoted as $\{(x_i, t_i)\}_{i \in [N]}$ with $[N]$ defined as the set $\{1, 2, \ldots, N\}$, CLIP aims to increase the similarity of the $(x_i, t_i)$ pair in the embedding space as follows. Let $I_i = E_I(x_i), T_i = E_T(t_i)$, CLIP maximizes the cosine similarity of the $(I_i, T_i)$ in the batch of $N$ pairs while minimizing the cosine similarity of $(I_i, T_j), j \neq i$ using a multi-class N-pair loss (Sohn, 2016; Radford et al., 2021). Once the image encoder $E_I$ and the text encoder $E_T$ are trained, CLIP can perform zero-shot classification for any set of labels: given a test image $x_1$, we can feed in the natural language names for a set of $M$ labels $\{t_j\}_{j \in [M]}$. The predicted label of $x_1$ is the label $t_k$ that has the largest cosine similarity among the embedding pairs: $(I_1, T_k)$.

## 3 METHOD

In this section, we describe CLIP-Dissect, an automatic, flexible and generalizable neuron labeling method for vision networks from popular convolutional neural networks (CNNs) to SOTA vision transformers (ViT). An overview of CLIP-Dissect algorithm is illustrated in Figure 2 and described in detail in Sec 3.1. We then introduce and discuss a few theoretically inspired choices for similarity function in Sec 3.2. Finally in Sec 3.3 we discuss how our method can benefit from more powerful models in the future.

### 3.1 CLIP-DISSECT OVERVIEW

**Inputs & Outputs.** The CLIP-Dissect algorithm has 3 inputs: **(i)** DNN to be dissected/probed, denoted as $f(x)$, **(ii)** a set of probing images, denoted as $\mathcal{D}_{\text{probe}}$ where $|\mathcal{D}_{\text{probe}}| = N$, **(iii)** a set of concepts, denoted as $\mathcal{S}, |\mathcal{S}| = M$.

The output of CLIP-Dissect is the neuron labels, which identify the concept associated with each individual neuron. Compared with Network Dissection, our goals are the same – we both want to inspect and detect concepts associated with each neuron. The input **(i)** is also the same, as we both want to dissect the DNN $f(x)$; however, the inputs **(ii)** and **(iii)** have differences. Specifically, in CLIP-Dissect, our $\mathcal{D}_{\text{probe}}$ does not require any concept labels and thus can be any publicly available dataset such as CIFAR-100, ImageNet, a combination of datasets or even unlabeled images collected from the internet. On the other hand, Network Dissection (Bau et al., 2017) can only use a $\mathcal{D}_{\text{probe}}$ that has been densely labeled with the concepts from the concept set $\mathcal{S}$. As a result, users of Network Dissection are limited to the $\mathcal{D}_{\text{probe}} = \mathcal{D}_{\text{Broden}}$ and the fixed concept set $\mathcal{S}$ of Broden unless they are willing to create their own densely labeled dataset. In contrast, the concept set $\mathcal{S}$ and probing dataset $\mathcal{D}_{\text{probe}}$ in our framework are *decoupled* – we can use any text corpus to form the concept set $\mathcal{S}$ and any image dataset $\mathcal{D}_{\text{probe}}$ independent of $\mathcal{S}$ in CLIP-Dissect, which significantly increases the flexibility and efficiency to detect neuron concepts.

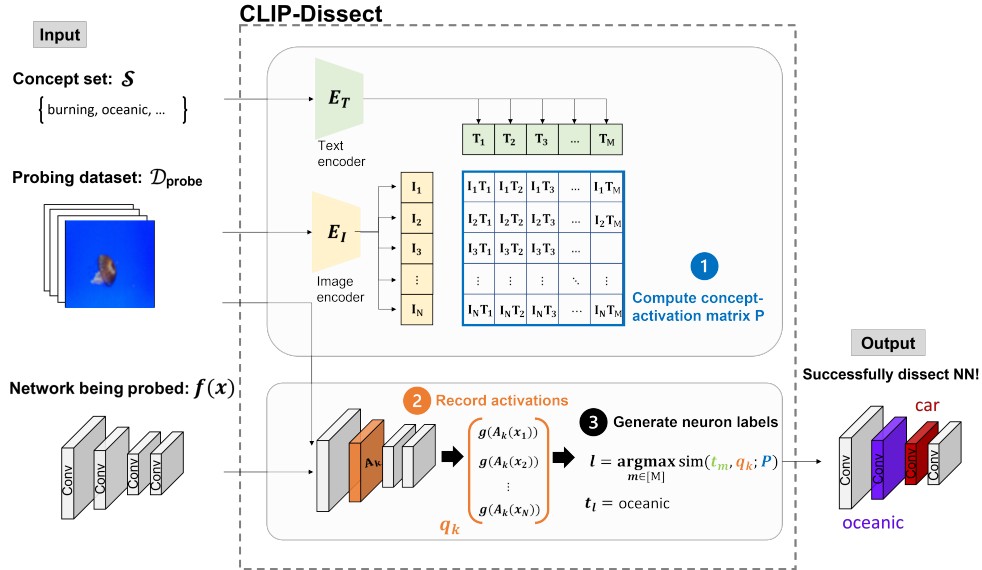

Figure 2: Overview of CLIP-Dissect: a 3-step algorithm to dissect neural network of interest

**Algorithm.** There are 3 key steps in CLIP-Dissect:

1. *Compute the concept-activation matrix $P$.* Using the image encoder $E_I$ and text encoder $E_T$ of a CLIP model, we first compute the text embedding $T_i$ of the concepts $t_i$ in the concept set $\mathcal{S}$ and the image embedding $I_i$ of the probing images $x_i$ in the probing dataset $\mathcal{D}_{\text{probe}}$. Next, we calculate the concept-activation matrix $P \in \mathbb{R}^{N \times M}$ whose $(i, j)$-th element is the inner product $I_i \cdot T_j$, i.e. $P_{i,j} = I_i \cdot T_j$.

2. *Record activations of target neurons.* Given a neuron unit $k$, compute the activation $A_k(x_i)$ of the $k$-th neuron for every image $x_i \in \mathcal{D}_{\text{probe}}$. Define a summary function $g$, which takes the activation map $A_k(x_i)$ as input and returns a real number. Here we let $g$ be the mean function that computes the mean of the activation map over spatial dimensions, but $g$ can be any general scalar function. We record $g(A_k(x_i))$, for all $i, k$.

3. *Determine the neuron labels.* Given a neuron unit $k$, the concept label for $k$ is determined by calculating the most similar concept $t_m$ with respect to its activation vector $q_k = [g(A_k(x_1)), \ldots, g(A_k(x_N))]^\top$, $q_k \in \mathbb{R}^N$. The similarity function $\texttt{sim}$ is defined as $\texttt{sim}(t_m, q_k; P)$. In other words, the label of neuron $k$ is $t_l$, where $l = \arg\max_m \texttt{sim}(t_m, q_k; P)$. Below we discuss different ways to define $\texttt{sim}$.

## 3.2 SIMILARITY FUNCTION

There are many ways to design the similarity function $\texttt{sim}$, and this choice has a large effect on the performance of our method. In particular, simple functions like cosine similarity perform poorly, likely because they place too much weight on the inputs that don't activate the neuron highly. We focus on the following 4 similarity functions and compare their results in the Table 3:

- **Cos.** Cosine similarity between the activation vector $(q_k)$ of the target neuron $k$ and the concept activation matrix $P_{:,m}$ from CLIP with the corresponding concept $t_m$:

$$\texttt{sim}(t_m, q_k; P) \triangleq \frac{P_{:,m}^\top q_k}{||P_{:,m}|| \cdot ||q_k||} \tag{1}$$

- **Rank reorder.** This function calculates the similarity between $q_k$ and $P_{:,m}$ by creating a vector $q_k'$, which has the values of $q_k$ in the order of $P_{:,m}$. I.e. $q_k'$ is generated by reordering the elements of $q_k$ according to the ranks of the elements in $P_{:,m}$. The full similarity function is defined below, and is maximized when the $q_k$ and $P_{:,m}$ have the same order:

$$\texttt{sim}(t_m, q_k; P) \triangleq -||q_k' - q_k||_p \tag{2}$$

- **WPMI** (**W**eighted **P**ointwise **M**utual **I**nformation). A mathematically grounded idea to derive `sim` based on mutual information as used in (Wang et al., 2020), where the label of a neuron is defined as the concept that maximizes the mutual information between the set of most highly activated images on neuron $k$, denoted as $B_k$, and the concept $t_m$. Specifically:

$$\mathrm{sim}(t_m, q_k; P) \triangleq \mathrm{wpmi}(t_m, q_k) = \log p(t_m|B_k) - \lambda \log p(t_m), \qquad (3)$$

where $p(t_m|B_k) = \Pi_{x_i \in B_k} p(t_m|x_i)$ and $\lambda$ is a hyperparameter.

- **SoftWPMI.** Finally, we propose a generalization of WPMI where we use the probability $p(x \in B_k)$ to denote the chance an image $x$ belongs to the example set $B_k$. Standard WPMI corresponds to the case where $p(x \in B_k)$ is either 0 or 1 for all $x \in \mathcal{D}_{\mathrm{probe}}$ while SoftWPMI relaxes the binary setting of $p(x \in B_k)$ to real values between 0 and 1.
This gives us the following function:

$$\mathrm{sim}(t_m, q_k; P) \triangleq \mathrm{soft\_wpmi}(t_m, q_k) = \log \mathbb{E}[p(t_m|B_k)] - \lambda \log p(t_m) \qquad (4)$$

where we compute $\log \mathbb{E}[p(t_m|B_k)] = \log(\Pi_{x \in \mathcal{D}_{\mathrm{probe}}}[1 + p(x \in B_k)(p(t_m|x) - 1)])$. As shown in our experiments (Table 3), we found SoftWPMI give the best results among the four and thus we use it in all our experiments unless otherwise mentioned.

Due to page constraint, we leave the derivation and details on how to calculate WPMI and SoftWPMI using only CLIP products matrix $P$, as well as our hyperparameter choices to Appendix A.1.

### 3.3 COMPABILITY WITH FUTURE MODELS

The current version of our algorithm relies on the CLIP (Radford et al., 2021) multimodal model. However, this doesn't have to be the case, and developing improved CLIP-like models has received a lot of attention recently, with many recent works reporting better results with an architecture similar to CLIP (Yu et al., 2022; Yuan et al., 2021; Zhai et al., 2022; Pham et al., 2021). If these models are released publicly, we can directly replace CLIP with a better model without any changes to our algorithm. As a result, our method will improve over time as general ML models get more powerful, while existing works (Bau et al., 2017; Hernandez et al., 2022) can't really be improved without collecting a new dataset specifically for that purpose. Similar to ours, the segmentation version of Network Dissection (Bau et al., 2020) can also be improved by using better segmentation models, but each improved segmentation model will likely work well for only a few tasks.

### 4 EXPERIMENTS

In this section, we provide both qualitative and quantitative results of CLIP-Dissect in Sec 4.1 and 4.2 respectively. We also provide an ablation study on the choice of similarity function in Sec 4.3 and compare computation efficiency in Sec 4.4. Finally, we show that CLIP-Dissect can detect concepts that do not appear in the probing images in Sec 4.5. We evaluate our method through analyzing two pre-trained networks: ResNet-50 (He et al., 2016) trained on ImageNet (Deng et al., 2009), and ResNet-18 trained on Places-365 (Zhou et al., 2017). Our method can also be applied to modern architectures such as Vision Transformers as discussed in Appendix A.5. Unless otherwise mentioned we use 20,000 most common English words[2] as the concept set $\mathcal{S}$.

Due to the page limit, we leave additional 9 experimental results in the Appendix. Specifically, Appendix A.2 shows additional qualitiative results discussed in Section 4.1. Appendix A.3 showcases our ability to detect low-level concepts but also discusses some limitations, such as sometimes outputting higher level concepts than warranted. Appendix A.4 shows how our method can be applied to generate compositional concepts, and Appendix A.5 shows that our method can be applied to Vision Transformer architecture and provides qualitative results. In Appendix A.6 we experiment with another potential method to measure quality of neuron explanations and show it also favors CLIP-Dissect. Appendix A.7 discusses the limitations of only displaying top-5 images for qualitative evaluations and showcases a wider range for some neurons. In Appendix A.8 we discuss how our method can be used to decide whether a neuron is interpretable or not. Appendix A.9 shows the qualitative effect of different similarity functions. Finally, in Appendix A.10 we evaluated our description quality for 500 randomly chosen neurons, and found descriptions generated by CLIP-Dissect to be a good match for 65.5% of the neurons on average.

---

[2]Source: https://github.com/first20hours/google-10000-english/blob/master/20k.txt

### 4.1 QUALITATIVE RESULTS

Figure 1 shows examples of descriptions for randomly chosen hidden neurons in different layers generated by CLIP-Dissect and the two baselines: Network Dissection (Bau et al., 2017) and MI-LAN (Hernandez et al., 2022). We do not compare against Compositional Explanations (Mu & Andreas, 2020) as it is much more computationally expensive (at least 200 times slower) and complementary to our approach as their composition could also be applied to our explanations. We observe that not every neuron corresponds to a clear concept and our method can detect low-level concepts on early layers and provide more descriptive labels than existing methods in later layers, such as the 'graduating' and 'nursery' neurons. These results use the union of ImageNet validation set and Broden as $D_{probe}$. In general we observe that MILAN sometimes gives very accurate descriptions but often produces descriptions that are too generic or even semantically incorrect (highlighted as red labels), while Network Dissection is good at detecting low level concepts but fails on concepts missing from its dataset. We compared against two versions of MILAN: MILAN(b) was trained to describe neurons of networks trained on "both" ImageNet and Places365, and MILAN(p) was only trained on Places365 neurons to test its generalization ability. Additional qualitative comparisons for interpretable neurons are shown in Figures 6 and 7 in Appendix A.2.

### 4.2 QUANTITATIVE RESULTS

Besides the qualitative comparison, in this section we propose the first quantitative evaluation to compare our methods's performance with baselines. The key idea is to compare the neuron labels generated for neurons where we have access to the *ground truth* descriptions – i.e. the final layer of a network, as the ground truth concept of the output layer neuron is the name of the corresponding class (class label). This allow us to objectively evaluate the quality of the generated neuron labels, which avoids the need for human evaluation and uses real function of the target neurons while human evaluations are usually limited to describing a few most highly activating images. We propose below two metrics for measuring the quality of explanations:

a) **Cos similarity:** We measure the cosine similarity in a sentence embedding space between the ground truth class name for the neuron (e.g. "sea lion" in Fig 3) and the explanation generated by the method. For embeddings, we use two different encoders: the CLIP ViT-B/16 text encoder (denoted as CLIP cos) and the all-mpnet-base-v2 sentence encoder (denoted as mpnet cos). See Figure 3 for an example of the similarity scores for descriptions of a single neuron.

b) **Accuracy:** We compute accuracy for a method as the percentage of neurons that the method assigns the exact correct label i.e. the class name. Note that we only measure accuracy in situations where the method chooses from a concept set that includes the exact correct label, such as Network Dissection for models trained on Places365 (not for ImageNet models since ImageNet labels are missing from Broden). We also did not measure accuracy of MILAN as MILAN generates explanations without a concept set and thus is unlikely to match the exact wording of the class name.

Table 1: The cosine similarity of predicted labels compared to ground truth labels on final layer neurons of ResNet-50 trained on ImageNet. The higher similarity the better. We can see that our method performs better when $D_{probe}$ and concept set are larger and/or more similar to training data.

| Method | $D_{probe}$ | Concept set $\mathcal{S}$ | CLIP cos | mpnet cos |
|---|---|---|---|---|
| Network Dissection (baseline) | Broden | Broden | 0.6929 | 0.2952 |
| MILAN(b) (baseline) | ImageNet val | - | 0.7080 | 0.2788 |
| CLIP-Dissect (Ours) | ImageNet val | Broden | 0.7393 | 0.4201 |
| | ImageNet val | 3k | 0.7456 | 0.4161 |
| | ImageNet val | 10k | 0.7661 | 0.4696 |
| | ImageNet val | 20k | 0.7900 | 0.5257 |
| | ImageNet val | ImageNet | 0.9766 | 0.9458 |
| | CIFAR100 train | 20k | 0.7300 | 0.3664 |
| | Broden | 20k | 0.7407 | 0.3945 |
| | ImageNet val | 20k | 0.7900 | 0.5257 |
| | ImageNet val + Broden | 20k | 0.7900 | 0.5233 |

Table 2: Performance when labeling final layer neurons of a ResNet18 trained on Places365. Accuracy measured on 267/365 neurons whose label is a directly included in Broden labels.

| Method | $D_{probe}$ | Concept set $\mathcal{S}$ | gt label annotation | Top1 Acc | CLIP cos | mpnet cos |
|---|---|---|---|---|---|---|
| Net-Dissect (baseline) | Broden | Broden | Yes | 43.82% | 0.8887 | 0.6697 |
| CLIP-Dissect (ours) | Broden | Broden | No | **58.05%** | **0.9106** | **0.7024** |

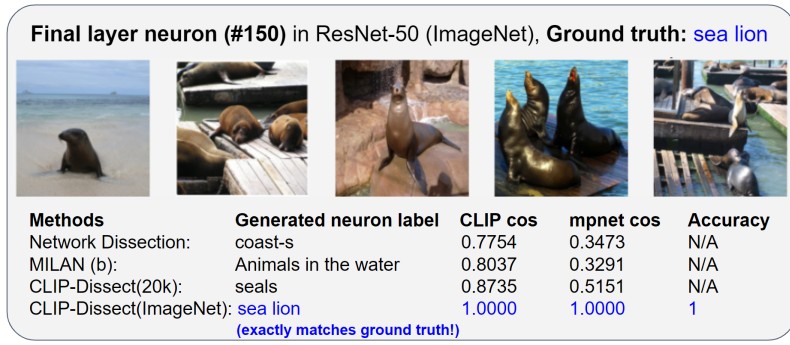

Figure 3: Example of a final layer neuron: we compare the descriptions generated by different methods and our metrics. Accuracy only evaluated for CLIP-Dissect with ImageNet labels as concept set since it is the only method where exact correct answer is a possible choice and therefore accuracy makes sense.

In Table 1, we can see that the labels generated by our method are closer to ground truth in sentence embedding spaces than those of Network Dissection or MILAN regardless of our choice of $D_{probe}$ or concept set $\mathcal{S}$. We can also see that using a larger concept set (e.g. 3k v.s. 20k) improves the performance of our method. Table 2 shows that our method outperforms Network Dissection even though this task is favorable to their method (as the Places365 dataset has large overlaps with Broden). We highlight that CLIP-Dissect can reach higher accuracy even though Network Dissection has access to and relies on the ground truth labels in Broden while ours does not.

### 4.3 CHOICE OF SIMILARITY FUNCTION

Table 3 compares the performance of different similarity functions used in CLIP-Dissect. We use accuracy and cos similarity in embedding space as defined in Sec 4.2 to measure the quality of descriptions. We observed that SoftWPMI performs the best and thus it is used in all other experiments unless otherwise mentioned. The effect of similarity function is shown qualitatively in Appendix A.9. Table 3 also showcases how CLIP-Dissect can give final layer neurons the correct label with a very impressive 95% accuracy.

Table 3: Comparison of the performance between similarity functions. We look at the final layer of ResNet-50 trained on ImageNet (same as Tab 1). We use $\mathcal{S} = 20k$ for cosine similarity and $\mathcal{S} =$ ImageNet classes for top1 accuracy. We can see SoftPMI performs best overall.

| Metric | Similarity function | CIFAR100 train | Broden | ImageNet val | ImageNet val + Broden | Average |
|---|---|---|---|---|---|---|
| mpnet cos similarity | cos | 0.2761 | 0.215 | 0.2823 | 0.2584 | 0.2580 |
| | Rank reorder | 0.3250 | 0.3857 | 0.4901 | 0.5040 | 0.4262 |
| | WPMI | 0.3460 | 0.3878 | **0.5302** | **0.5267** | 0.4477 |
| | SoftWPMI | **0.3664** | **0.3945** | 0.5257 | 0.5233 | **0.4525** |
| Top1 accuracy | cos | 8.50% | 5.70% | 15.90% | 11.40% | 10.38% |
| | Rank reorder | 36.30% | 57.50% | 89.80% | 89.90% | 68.38% |
| | WPMI | 23.80% | 47.10% | 87.00% | 86.90% | 61.20% |
| | SoftWPMI | **46.20%** | **70.50%** | **95.00%** | **95.40%** | **76.78%** |

## 4.4 COMPUTATIONAL EFFICIENCY

Table 4 shows the runtime of different automated neuron labeling methods when tasked to label all the neurons of five layers in ResNet-50. We can see our method runs in just 4 minutes, more than 10, 60 and 200+ times faster than the baselines MILAN (Hernandez et al., 2022), Network Dissection (Bau et al., 2017) and Compositional Explanations (Mu & Andreas, 2020) respectively.

Table 4: The time it takes to describe the layers ['conv1', 'layer1', 'layer2', 'layer3', 'layer4'] of ResNet-50 via different methods using our hardware(Tesla P100 GPU).We can see CLIP-Dissect is much more computationally efficient than existing methods.

| Method | CLIP-Dissect | Network Dissection | Compositional Explanations | MILAN |
|---|---|---|---|---|
| Runtime | **3min50s** | >4 hrs | >>14 hours | 55min 30s |

## 4.5 DETECTING CONCEPTS MISSING FROM $D_{probe}$

One surprising ability we found is that our method is able to assign the correct label to a neuron even if $D_{probe}$ does not have any images corresponding to that concept. For example, CLIP-Dissect was able to assign the correct dog breed to 46 out of 118 neurons detecting dog breeds, and correct bird species to 22 out of 59 final layer neurons of ResNet-50 trained on ImageNet, while using CIFAR-100 training set as $D_{probe}$, which doesn't include any images of dogs or birds. This is impossible for any label based methods like NetDissect (Bau et al., 2017) and Compositional Explanations (Mu & Andreas, 2020) (as IoU will be 0 for any concept not in $D_{probe}$), and unlikely for methods based on captioning highly activated images like MILAN (Hernandez et al., 2022) (as humans won't assign a caption missing from activated images). Example labels and highest activating probe images can be seen in Figure 4.

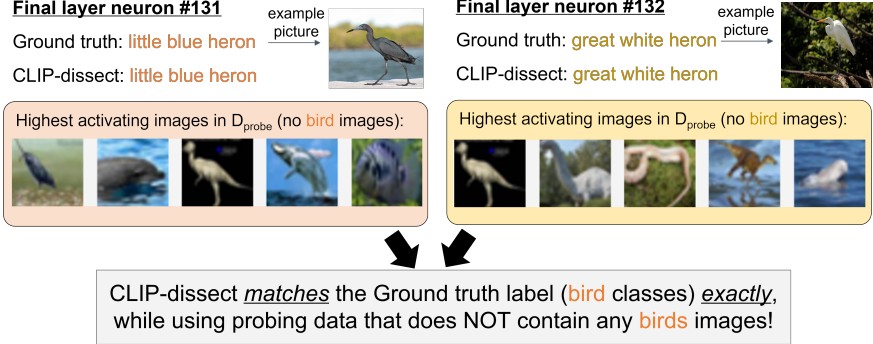

Figure 4: Example of CLIP-Dissect correctly labeling neurons that detect the little blue heron and the great white heron based on pictures of dolphins and dinosaurs in CIFAR. CIFAR100 does not contain any bird images but CLIP-Dissect can still get correct concept.

## 5 USE CASE OF CLIP-DISSECT

In this section, we present a simple experiment to showcase how we can use CLIP-Dissect to gain new insights on neural networks. By inspecting the ResNet-50 network trained on ImageNet with CLIP-Dissect, we discover the following phenomenon and evidence for it: **the higher the weight between two neurons, the more similar concepts they encode**, as shown in Figure 5. This makes sense since a high positive weight causally makes the neurons activate more similarly, but the extent of this correlation is much larger than we expected, as each final layer neuron has 2048 incoming weights so we would not expect any single weight to have that high of an influence. A consequence of the similarity in concepts is that the second-to-last layer already encodes quite complete representations of certain final layer classes in individual neurons, instead of the representation for these

classes being spread across multiple neurons. For example Fig 5a shows that the 3 neurons with highest outgoing weights already seem to be accurately detecting the final layer concept/class label they're connected to.

To make these results more quantitative, in Figure 5b we measure the similarity of concepts encoded by the neurons connected via highest weights in the final layer of ResNet-50. For layer4 neurons, we used CLIP-Dissect to determine their concept, while for the final layer neurons we used the ground truth i.e. class label in text form. We can clearly see that higher weights connect more similar concepts together, and the average similarity decreases exponentially as a function of $k$ when averaging similarities of neurons connected via the top $k$ weights. To further test this relationship, we found that the mpnet cos similarity between concepts encoded by two neurons and the weight connecting them are correlated with $r = 0.120$ and p-value $< 10^{-300}$ (probability of no correlation is practically 0) when calculated over all 2 million weights in the final layer. If we only look at the highest 50000 weights, the correlation is even higher with $r = 0.258$, p-value $< 10^{-300}$.

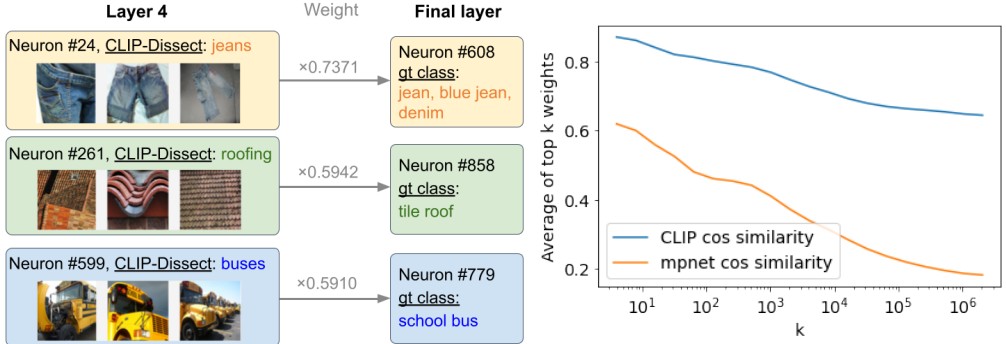

(a) Visualization of 3 highest weights of final layer.   (b) Average cosine similarity between concepts.

Figure 5: a) 3 highest weights of the final layer of ResNet-50 trained on ImageNet, we can see neurons connected by the highest weights are detecting very much the same concept. b) Cosine similarities between the concepts of neurons connected by highest weights. The higher the weight between neurons, the more similar a concept they represent.

## 6 LIMITATIONS AND CONCLUSIONS

**Limitations:** The main limitation of our method compared to previous work is that it's not taking advantage of the spatial information of neuron activations. This causes some difficulties in detecting lower level concepts, but we are still able to detect many low level/localized patterns as discussed in Section A.3. Secondly, our method currently works well only on concepts and images that CLIP works well on, and while this already covers a larger set of tasks than what existing neuron labeling methods perform well on, CLIP-Dissect may not work out of the box on networks trained on tasks that require highly specific knowledge such as classifying astronomical images. However, our method is compatible with future large vision-language models as long as they share a similar structure to CLIP, and CLIP-like models trained for a specific target domain. Finally, not all neurons can be described well by simple terms such as single word explanations. While we can augment the space of descriptions using a different concept set $\mathcal{S}$, or creating compositional explanations as discussed in Appendix A.4, some neurons may have a very complicated function or perform different functions at different activation ranges. For the most part, current methods including ours will be unable to capture this full picture of complicated neuron functions.

**Conclusions:** In this work, we have developed CLIP-Dissect, a novel, flexible and computationally efficient framework for automatically identifying concepts of hidden layer neurons. We also proposed new methods to quantitatively compare neuron labeling methods, which is based on labeling final layer neurons. We have shown CLIP-Dissect can match or outperform previous automated labeling methods both qualitatively and quantitatively and can even detect concepts missing from the probing dataset. Finally we used CLIP-Dissect to discover that neurons connected by a high weight often represent very similar concepts.

ACKNOWLEDGEMENT

The authors would like to thank anonymous reviewers for valuable feedback to improve the manuscript. The authors also thank MIT-IBM Watson AI lab for computing support in this work. T. Oikarinen and T.-W. Weng are supported by National Science Foundation under Grant No. 2107189.

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

## A APPENDIX

### A.1 SIMILARITY FUNCTION DETAILS AND DERIVATION

**Rank reorder hyperparameters:**

The results of Table 3 are using top 5% of most highly activating images for each neuron and using $p = 3$ for the $l_p$-norm.

**WPMI:**

In this section, we show that one choice of similarity function $\text{sim}(t_m, q_k; P)$ can be derived based on the weighted point-wise mutual information (wpmi). Note that wpmi is also used in (Hernandez et al., 2022) but in a different way – our approach can compute wpmi directly from the CLIP products $P$ and does not require any training, while (Hernandez et al., 2022) train two models to estimate wpmi.

To start with, by definition, the wpmi between a concept $t_m$ and the most highly activated images $B_k$ of neuron $k$ can be written as

$$\text{wpmi}(t_m, q_k) = \log p(t_m|B_k) - \lambda \log p(t_m) \tag{5}$$

Here $B_k$ is the set of images that most highly activates neuron $k$, i.e. the top indices of $q_k$. First we can compute $p(t_m|x_i) = \text{softmax}(aP_{i,:})_m$, where $\text{softmax}(z)_n = \frac{e^{z_n}}{\sum_{j=1}^{N} e^{z_j}}$ with $z \in \mathbb{R}^N$, $P_{i,:}$ is the $i$-th row vector of the concept-activation matrix $P$ and $a$ is a scalar temperature constant. This is the probability that CLIP assigns to a concept $t_m$ for image $x_i$ when used as a classifier.

We then define $p(t_m|B_k)$ as the probability that all images in $B_k$ have the concept $t_m$, which gives us $p(t_m|B_k) = \Pi_{x_i \in B_k} p(t_m|x_i)$. Thus, we have

$$\log p(t_m|B_k) = \sum_{x_i \in B_k} \log p(t_m|x_i) \tag{6}$$

which is the 1st term in Eq (5). Next, we can approximate the 2nd term $p(t_m)$ in Eq (5) as follows: $p(t_m)$ is the probability that a random set of images $B$ will be described by $t_m$. Since we don't know the true distribution for a set of images, an efficient way to approximate this is to average the probability of $t_m$ over the different neurons we are probing. This can be described by the following equation:

$$p(t_m) = \mathbb{E}_B[p(t_m|B)] \approx \frac{\sum_{j \in C} p(t_m|B_j)}{|C|} = \frac{\sum_{j \in C} \Pi_{x_i \in B_j} p(t_m|x_i)}{|C|} \tag{7}$$

where $C$ is the set of neurons in the layer we are probing. Thus we can plug Eq. (6) and Eq. (7) in to Eq. (5) to compute wpmi through the CLIP model:

$$\text{wpmi}(t_m, q_k) = \sum_{x_i \in B_k} \log p(t_m|x_i) - \lambda \log \left( \sum_{j \in C} \Pi_{x_i \in B_j} p(t_m|x_i) \right) + \lambda \log |C| \tag{8}$$

So we can use the above Eq (8) in our CLIP-Dissect and set $\text{sim}(t_m, q_k; P) = \text{wpmi}(t_m, q_k)$ in the algorithm.

For our experiments we use $a = 2$, $\lambda = 0.6$ and top 28 most highly activating images for neuron $k$ as $B_k$ which were found to give best quantitave results when describing final layer neurons of ResNet-50.

**SoftWPMI:**

SoftWPMI is an extension of wpmi as defined by Eq. (8) into settings where we have uncertainty over which images should be included in the example set $B_k$. In WPMI the size of example set is defined beforehand, but it is not clear how many images should be included, and this could vary from neuron to neuron. In this description, we assume that there exists a true $B_k$ which includes images from $D_{probe}$ if and only if they represent the concept of neuron $k$. We then define binary indicator random variables $X_i^k = \mathbb{1}[x_i \in B_k]$ which take value 1 if the ith image is is in set the $B_k$, and we define $X^k = \{X_1^k, ..., X_M^k\}$.

Our derivation begins from the observation that we can rewrite $p(t_m|B_k)$ from above as:

$$p(t_m|B_k) = \Pi_{x_i \in B_k} p(t_m|x_i) = \Pi_{x_i \in D_{probe}} p(t_m|x_i)^{\mathbb{1}[x_i \in B_k]} = \Pi_{x_i \in D_{probe}} p(t_m|x_i)^{X_i^k} \quad (9)$$

Now:

$$\mathbb{E}_{X_i^k}[p(t_m|x_i)^{X_i^k}] = p(x_i \in B_k)p(t_m|x_i) + (1 - p(x_i \in B_k)) = 1 + p(x_i \in B_k)(p(t_m|x_i) - 1) \quad (10)$$

If we assume the $X_i^k$ are statistically independent, we can write:

$$\mathbb{E}_{X^k}[p(t_m|B_k)] = \Pi_{x_i \in D_{probe}} \mathbb{E}_{X_i^k}[p(t_m|x_i)^{X_i^k}] = \Pi_{x_i \in D_{probe}}[1 + p(x_i \in B_k)(p(t_m|x_i) - 1)] \quad (11)$$

$$\Rightarrow \log \mathbb{E}_{X^k}[p(t_m|B_k)] = \sum_{x_i \in D_{probe}} \log(1 + p(x_i \in B_k)(p(t_m|x_i) - 1)) \quad (12)$$

Note Equation (10) goes to 1 if $p(x_i \in B_k) = 0$ (i.e. no effect in a product) and to $p(t_m|x_i)$ if $p(x_i \in B_k) = 1$. So Eq. (12) reduces to Eq. (6) of standard WPMI if $p(x_i \in B_k)$ is either 1 or 0 for all $x_i \in D_{probe}$. In other words, we are considering a "soft" membership in $B_k$ instead of "hard" membership of standard WPMI.

To get the second term for wpmi, $p(t_m)$, i.e. probability that text $t_m$ describes a random example set $B_k$, we can approximate it like we did in Eq. (7) by using the example sets for other neurons we are interested in.

$$p(t_m) = \mathbb{E}_{B_i}[\mathbb{E}_{X^i}[p(t_m|B_i)]] \approx \frac{\sum_{j \in C} \mathbb{E}_{X^j}[p(t_m|B_j)]}{|C|}$$

$$\rightarrow \frac{\sum_{j \in C} \mathbb{E}_{X^j}[p(t_m|B_j)]}{|C|} = \frac{\sum_{j \in C} \Pi_{x \in D_{probe}}[1 + p(x \in B_j)(p(t_m|x) - 1)]}{|C|} \quad (13)$$

Finally, we can compute full SoftWPMI with Eq. (12) and Eq. (13) and use it as similarity function in CLIP-Dissect:

$$\text{soft\_wpmi}(t_m, q_k) = \sum_{x_i \in D_{probe}} \log(1 + p(x_i \in B_k)(p(t_m|x_i) - 1))$$

$$-\lambda \log \left( \sum_{j \in C} \Pi_{x \in D_{probe}}[1 + p(x \in B_j)(p(t_m|x) - 1)] \right) + \lambda \log|C| \quad (14)$$

One thing we haven't yet discussed is the choice of $p(x \in B_k)$. There is flexibility and this probability could be derived from the activations of neuron $k$ on image $x$, by for example by taking a scaled sigmoid, or it could be based on the ranking of the image.

For our experiments we found ranking based probability to perform the best, and used $p(x \in B_k)$ linearly decreasing from 0.998 of the most highly activating image for neuron $k$ to 0.97 for 100th most highly activating image and 0 for all other images. Thus in practice we only have to use the 100 images when calculating SoftWPMI instead of full $D_{probe}$ which is much more computationally efficient. For other hyperparameters we used $a = 10$ and $\lambda = 1$.

## A.2 ADDITIONAL QUALITATIVE RESULTS

Additional visualization on ResNet-18 and ResNet-50 in Figs 6 and 7, continued from Section 4.1.

## Resnet-18(Places 365) Layer 4

**Most interpretable neurons
(Network Dissection)**

**Most interpretable neurons
(CLIP-Dissect)**

Figure 6: Explanations of most interpretable neurons in the second to last layer of ResNet-18 trained on Places365. Displayed together with 5 most highly activating images for that neuron. We have subjectively colored the descriptions green if they match these 5 images, yellow if they match but are too generic and red if they do not match. Both Network Dissection and CLIP-Dissect do very well while MILAN struggles to explain some neurons. MILAN(b) is trained on both ImageNet and Places365 networks, while MILAN(i) is only trained on ImageNet. Both MILAN networks perform similarly here but the ImageNet version misses/is too generic for more neurons, such as labeling a bus neuron as "vehicles". The neurons on the left have highest IoU according to MILAN while neurons on the right have highest similarity to the concept according to our similarity function.

## Resnet-50(ImageNet) Layer 4

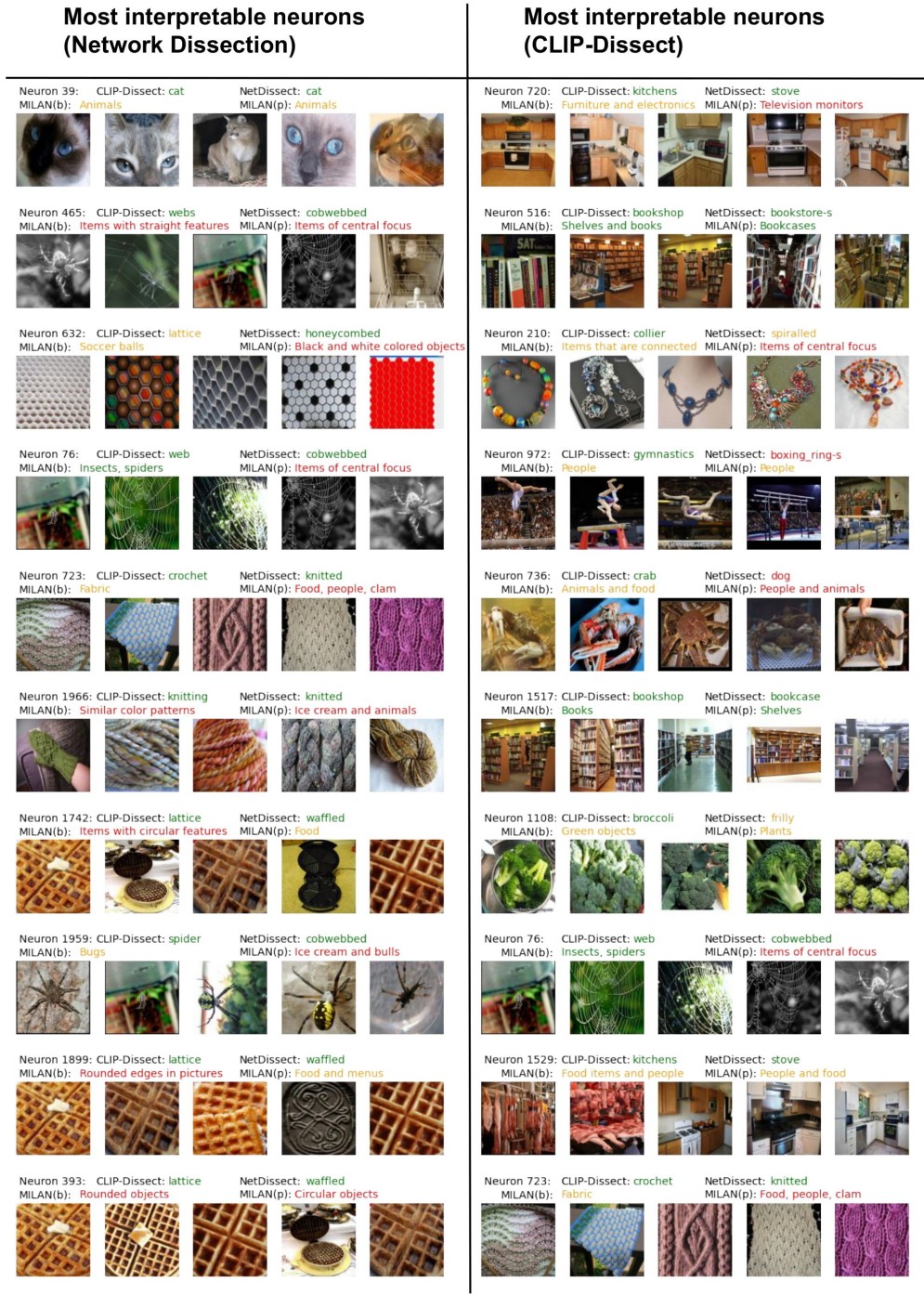

Figure 7: Explanations of most interpretable neurons in the second to last layer of ResNet-50 trained on ImageNet. Displayed together with 5 most highly activating images for that neuron. We have subjectively colored the descriptions green if they match these 5 images, yellow if they match but are too generic and red if they do not match. Both CLIP-Dissect and Network Dissection perform well on these most interpretable neurons except for a few failures by Network Dissection, while MILAN often gives concepts that are too generic. MILAN(b) is trained on both ImageNet and Places365 networks, while MILAN(p) is only trained on Places365. We can see the Places trained model is struggling more with concepts like spiders, indicating issues with generalization.

### A.3 LOW LEVEL CONCEPTS

In this section we show additional results of probing low level concepts, using two networks trained on ImageNet, ResNet-50 and ResNet-152. For ResNet-152 we also compare against human annotations for these neurons from MILAnnotations (Hernandez et al., 2022).

The results for ResNet-152 can be seen in Figure 8. We can see CLIP-Dissect is able to accurately detect many lower level concepts, such as colors in Conv1 neurons 1,3,10 and Layer1 neuron 4, as well as detecting that neuron 3 of Layer1 activates specifically for the *text/label*, without having access to the activation pattern, while MILAN fails to detect this.

However, CLIP-Dissect does also have some failure modes on lower level patterns.

- **Failure to differentiate between concept and correlated objects:** This leads to higher level outputs than desired. For example: CLIP-Dissect gives the concept *underwater* to conv1 neuron 16, while the true concept is probably more similar to *blue* (human annotators also made this mistake), or conv1 neuron 28 where CLIP-Dissect outputs *zebra* while the neuron is likely just detecting *stripes*. The worst example of this is neuron conv1 neuron 24, which simply activates on white background, but this is entirely missed by CLIP-Dissect as it's not good at focusing on the background.

- **Unintrepretable neurons:** Some neurons seem to be not interpretable, e.g. Conv1 neuron 2. CLIP-Dissect outputs *music* which seems incorrect, but neither human annotators or MILAN were able to assign a clear concept for the neuron either. See Appendix A.8 for more analysis on uninterpretable neurons.

Similar observations also hold for the ResNet-50 model as can be seen in Figure 9. Note we do not have human annotations to compare against for this network.

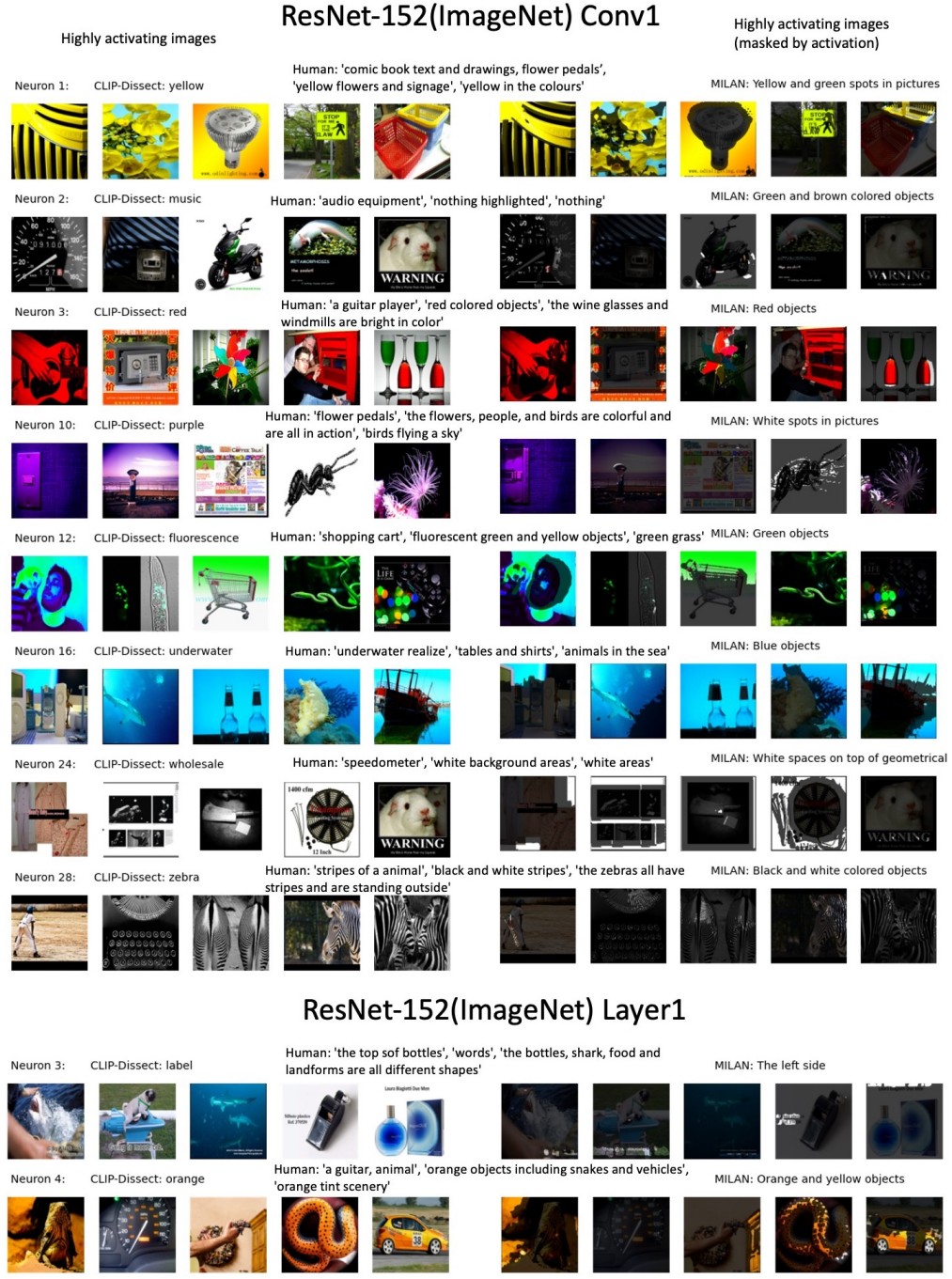

Figure 8: Descriptions for select neurons in early layers of ResNet-152, showcasing both the successes and failure modes of CLIP-Dissect. For this figure we used max as the summary function $g$ for CLIP-Dissect to be comparable to MILAnnotations. For evaluation we used the MILAN model trained on only Places models to avoid overfitting.

## ResNet-50 (ImageNet) Layer1

Figure 9: Descriptions of most interpretable neurons (highest similarity/IoU) of an early layer in ResNet-50.

## A.4 COMPOSITIONAL CONCEPTS

In the sections above our method has focused on choosing the most fitting concept from the pre-defined concept set. While changing the concept set in CLIP-Dissect is as easy as editing a text file, we show it can also detect more complex compositional concepts. We experimented with generating explanations by searching over text concatenations of two concepts in our concept space. To reduce computational constraints, we only looked at combinations of 100 most accurate single word labels for each neuron. Example results are shown in Fig 10. While the initial results are promising, some challenges remain to make these compositional explanations more computationally efficient and consistent, which is an important direction for future work.

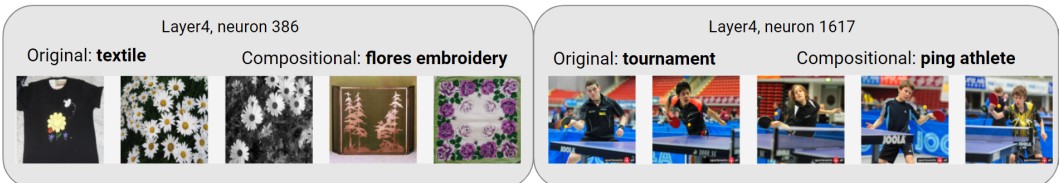

Figure 10: An example of compositional explanations generated by our method for two neurons of ResNet50 trained on ImageNet.

## A.5 VISION TRANSFORMER

Since our method does not rely on the specifics of CNNs in its operation, we can easily extend it to work on different architectures, such as the Vision Transformer, specifically ViT-B/16 (Dosovitskiy et al., 2020) model trained on ImageNet. We have visualized most interpretable neurons, their highly activating images and their descriptions in Figure 11. Interestingly, we found the highly interpretable neurons to be very location focused, i.e. *kitchens* or *highways* despite the network being trained on object level labels (ImageNet).

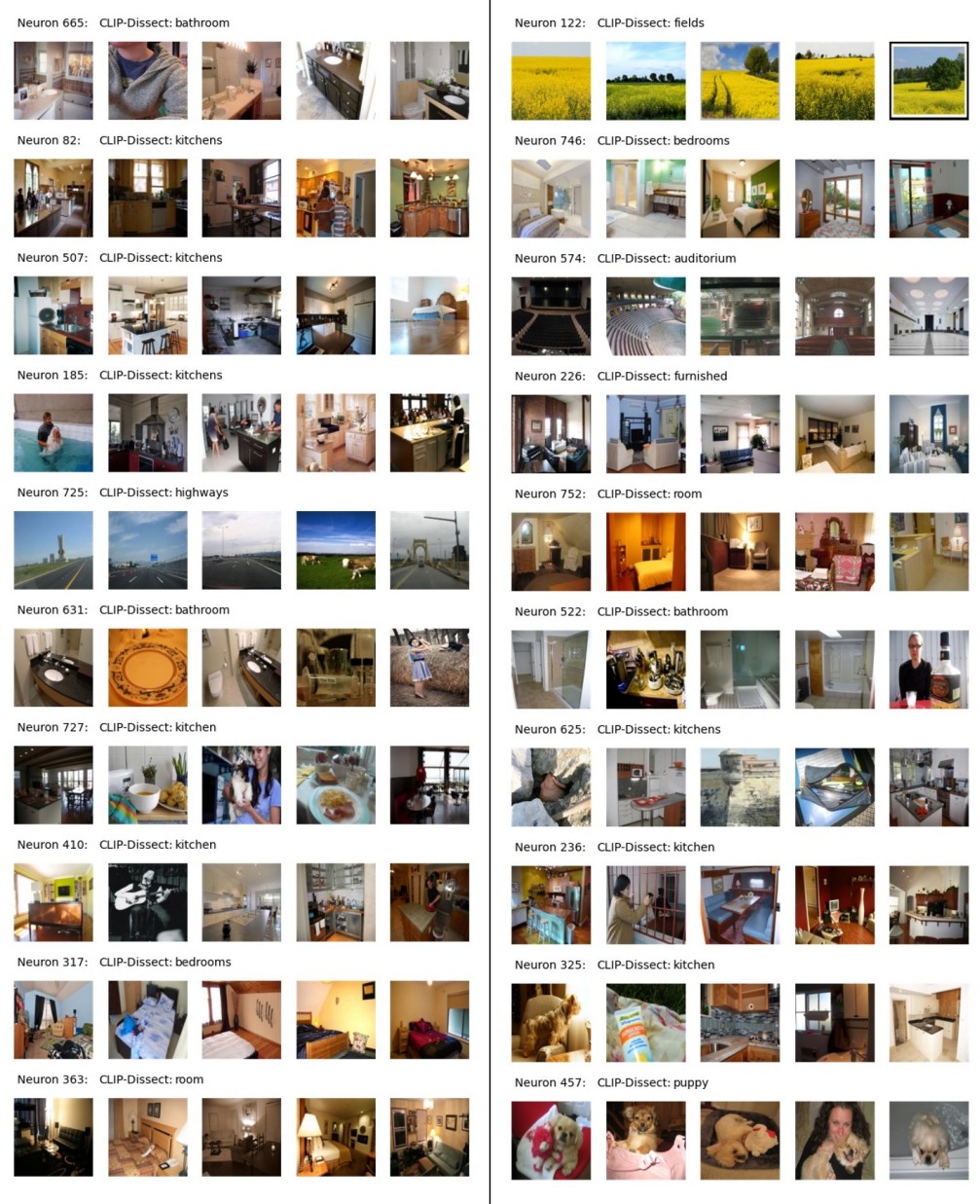

Figure 11: Most interpretable neurons after the 'encoder' module (one of the last layers) of a Vision Transformer.

### A.6 PREDICTING INPUT CLASS FROM NEURON DESCRIPTIONS

In this section we follow reviewer suggestion and study whether the class of the image can be predicted based on descriptions of highly activating neurons. We perform an experiment on the neurons in the second-to-last layer (last hidden layer) and study if the descriptions of highly activating neurons in this layer can be used to predict what class the input is from.

As we show in below experiments, the neuron description generated from CLIP-Dissect gives overall higher prediction accuracy than the neuron descriptions from prior work (Network Dissection and MILAN), which suggests our proposed method generates better neuron description than prior work. Below we outline the steps we used to predict the class of an image based on the internal neuron descriptions from different neuron labelling methods:

1. Following our notation in Sec 3, for an image $x_i$, record the activations of neurons in the second-to-last layer $g(A_j(x_i))$ (where $j$ denotes the neuron index) as well as the predicted class $c$.

2. Find the neuron with the highest positive contribution to the predicted class $c$, which can be computed as $k = \arg\max_j W_{c,j} \cdot g(A_j(x_i))$.

3. Obtain the description $t_k$ for neuron $k$ using an automated description method.

4. Find the class name that is most similar to the description of the highest contributing neuron, and predict this class as the images class. Similarity is measured using an average of cosine similarities in the sentence embedding space of CLIP text encoder and mpnet sentence embedding space discussed in section 4.2.

We performed this experiment on ResNet-50 trained on ImageNet, and found that predicting with the above algorithm and CLIP-Dissect neuron descriptions we were able to correctly predict the class of 10.28% of the images. In contrast, when we used descriptions from Network Dissection we were only able to predict the class of 3.36% of the images, and with MILAN(base) only 2.31% of the time. This gives evidence towards our CLIP-Dissect descriptions being higher quality than Network Dissection and MILAN. It's worth noting that overall we would not expect to reach a very high predictive accuracy using the above method as the most contributing neuron often does have a completely different role than the target class. However if we study the same network with different neuron description methods, we would expect a better neuron description method would return a higher prediction accuracy, assuming at least some of the important neurons indeed do have similar role to the class itself. This method has the benefit of being automated, going beyond visualizing few most highly activating images and being able to analyze hidden layer description quality. We think methods like this are an interesting future direction for evaluating neuron description methods.

### A.7 VISUALIZING WIDER RANGE OF ACTIVATIONS

So far our qualitative evaluation has focused on whether the description matches the 5 or 10 images in $\mathcal{D}_{\text{probe}}$ that activate the neuron the most. However this does not give a full picture of the function of the neuron, and in this section we explore how the neurons activate on a wider range of input images. In particular, in Figures 12 and 13 we visualize the most interpretable neurons of two layers of ResNet-18 (Places 365) and ResNet-50 (ImageNet) by uniformly sampling images from the top 0.1%, 1% 5% of most highly activating images. We can see that the descriptions tend to match quite well for the 0.1% of most highly activating images, but the top 1% and top 5% images start to be of quite different concepts only slightly related to the description. We also notice that low level concepts like colors tend to be more consistently represented in top 1% and 5% images while higher level concepts are not. This result is somewhat expected as the $\mathcal{D}_{\text{probe}}$ does not have that many images for each higher level concept, but highlights the need to explore neuron activations beyond just the few most highly activated images.

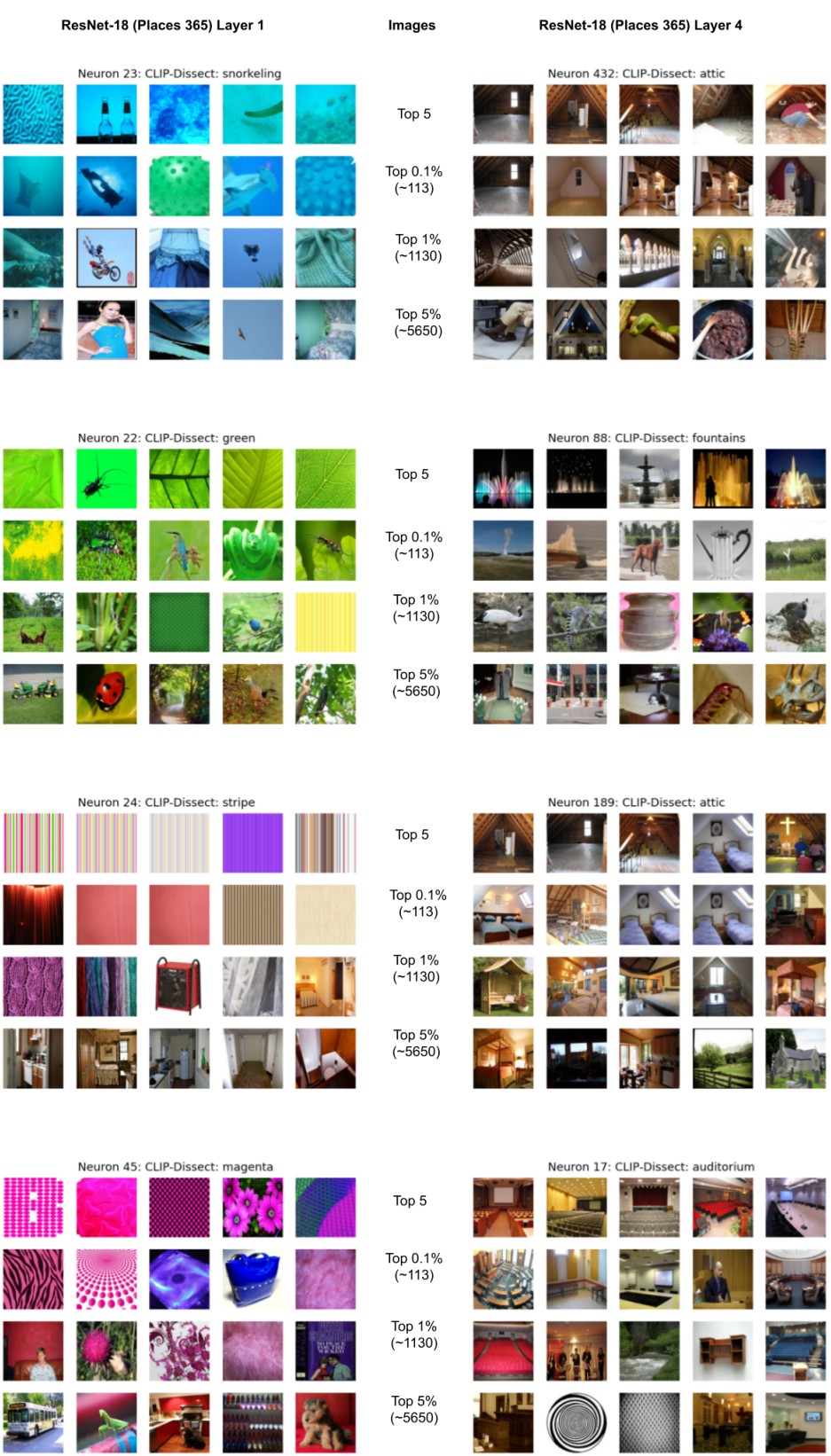

Figure 12: Most interpretable neurons of ResNet-18, showcasing randomly sampled images of a wide range of most highly activating images for that neuron.

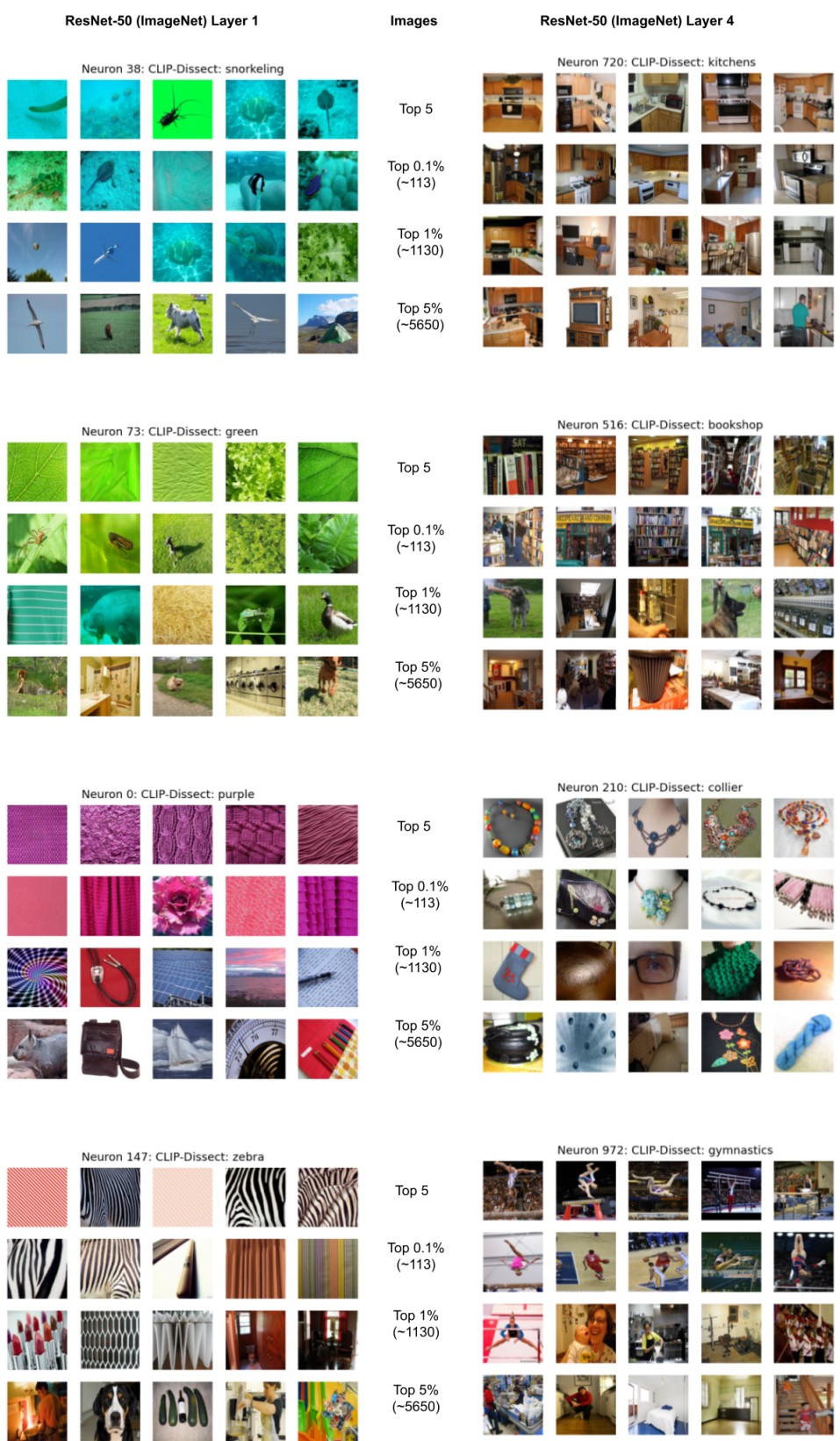

Figure 13: Most interpretable neurons of ResNet-50, showcasing randomly sampled images of a wide range of most highly activating images for that neuron.

### A.8 INTERPRETABILITY CUTOFF

Our method can also be used to quantify which neurons are 'interpretable'. Since each description for a neuron is associated with a similarity score, and the higher the similarity the more accurate the description, we consider a neuron $k$ with description $t$ as interpretable if $\text{sim}(t, q_k; P) > \tau$. To choose the best cutoff $\tau$, we leverage our experiment in section A.10. In particular, we choose the lowest $\tau$ such that interpretable neurons will have an average description score of 0.75 or higher (compared to 0.655 of all neurons). This gives us a cutoff threshold $\tau = 0.16$ with the proposed SoftWPMI similarity function. In contrast, the neurons with SoftWPMI $\leq \tau$ have an average description score of 0.5257, which is lower than that of interpretable neurons (0.75) and all neurons (0.655). Thus, it suggests that the similarity score of SoftWPMI is a useful indicator of description quality. Using this cutoff $\tau$, we find 69.7% of neurons in ResNet-18 (Places-365) and 77.8% of neurons in ResNet-50 (ImageNet) to be interpretable, indicating that around 20-30% of neurons do not have a simple explanation for their functionality, i.e. are 'uninterpretable'.

### A.9 QUALITATIVE EFFECT OF SIMILARITY FUNCTION

Figure 14 shows the descriptions generated by our CLIP-Dissect when using simple cosine similarity as the similarity function. As we can see the performance is very poor, only adequately describing one of the 8 neurons displayed, highlighting the need for more sophisticated similarity functions we introduced.

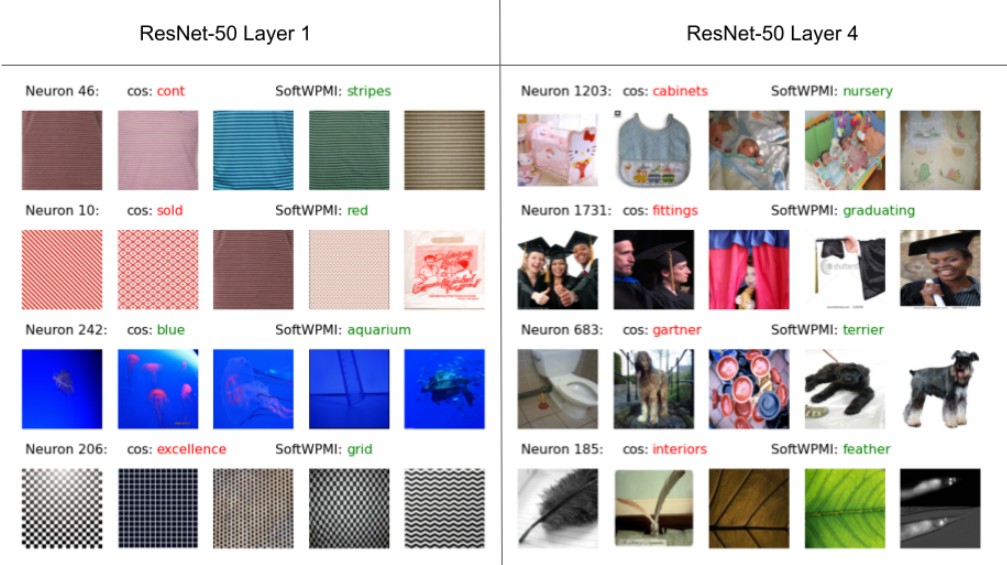

Figure 14: Explanations for same neurons as Figure 1 showcasing the qualitative difference between using a simple cos similarity as the similarity function for CLIP-Dissect vs our best performing SoftWPMI similarity function.

## A.10 LARGER SCALE EXPERIMENT ON DESCRIPTION QUALITY

In this section we perform a larger scale analysis of the neuron description quality provided by CLIP-Dissect. We evaluated the description quality of 50 randomly selected neurons for each of the 5 layers and 2 models studied, for a total of 1000 evaluations. Each evaluator was presented with 10 most highly activating images, and answered the question: "Does the description: '{}' match this set of images?" An example of the user interface is shown in Figure 15. Each evaluation had three options which we used with the following guidelines:

- Yes - Most of the 10 images are well described by this description

- Maybe - Around half (i.e. 3-6) of the images are well described, or most images are described relatively well (accurate but too generic, or slightly inaccurate)

- No - Most images are poorly described by this caption

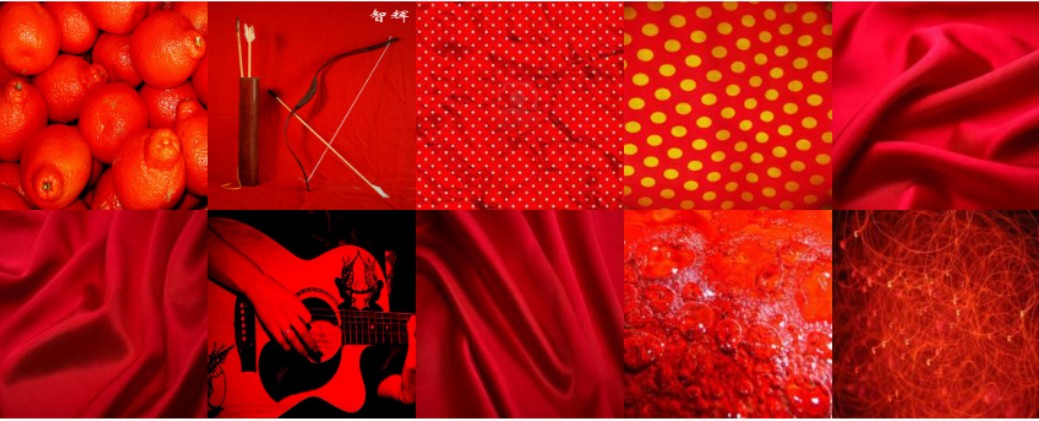

Question 1/50. Description: red

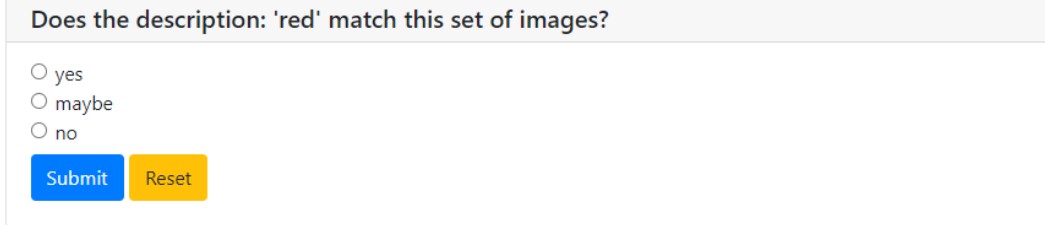

Figure 15: Example of the user interface for our evaluation of description quality.

These evaluations were turned into a numeric score with the following formula: yes:1, maybe:0.5, no:0. Table 5 shows the average description score across different neurons and evaluators for each of the layers evaluated. This average score can be thought of as the percentage of neurons well described. We observed that overall the descriptions are good for 55-80% of neurons depending on the layer, with the average score across all evaluations being 0.655. In addition we notice that the very early and very late layers are most interpretable, corresponding to clear low or high level concepts, while the middle layers seem to be harder to describe. It is worth noting that we are evaluating random neurons here, i.e. the neurons are selected randomly, so the displayed neuron may not be interpretable in the first place – in many cases when the description does not match are because the neuron itself is not 'interpretable', i.e. there is no simple description that corresponds to the neurons functionality.

Even with the evaluation guidelines described above, these evaluations are subjective, and we found that our two evaluators agreed on 68.4% of the neurons with the vast majority of disagreements being between yes/maybe or maybe/no, with only 2.4% of neurons having a yes from one evaluator and no from another. For transparency, we have included all 50 neurons, their descriptions and two

Table 5: The average description scores of the CLIP-Dissect descriptions for neurons of different layers. An average score of 1.0 indicates all descriptions match the neurons highly activating images, 0.0 means none do, and 0.5 could mean anything from all neurons 'maybe' match or half the neurons fully match and half don't at all.

| Model\Layer | conv1 | layer1 | layer2 | layer3 | layer4 |
|---|---|---|---|---|---|
| ResNet-18 (Places 365) | 0.805 | 0.635 | 0.635 | 0.410 | 0.695 |
| ResNet-50 (ImageNet) | 0.815 | 0.670 | 0.550 | 0.640 | 0.695 |

evaluator's evaluations (E1, E2) of these descriptions for ResNet-50 layer conv1 in Figures 16, 17 and 18 and for ResNet-50 layer4 in Figures 19, 20 and 21

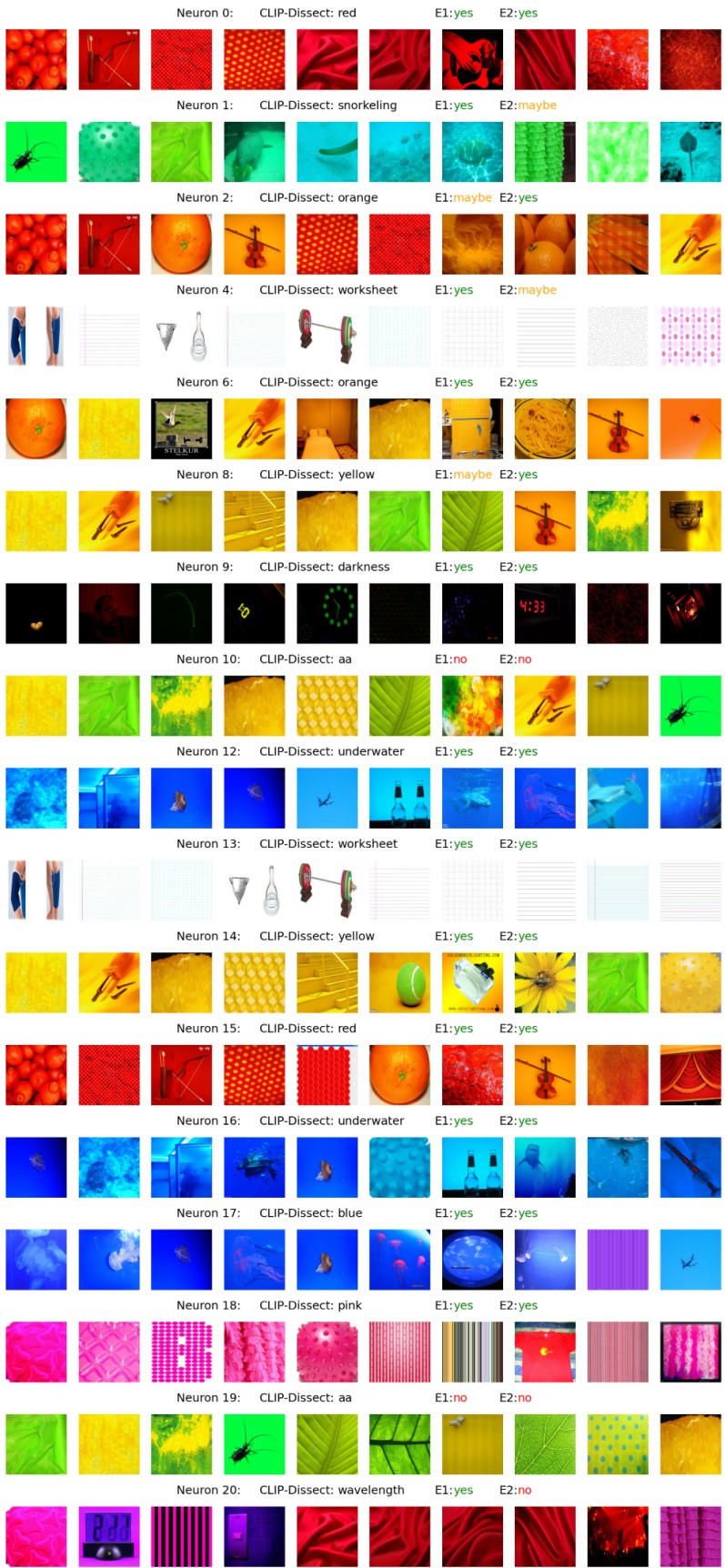

Figure 16: Random neurons of 'conv1' in ResNet-50 (ImageNet), and E1, E2's evaluation on the description of CLIP-Dissect (PART 1).

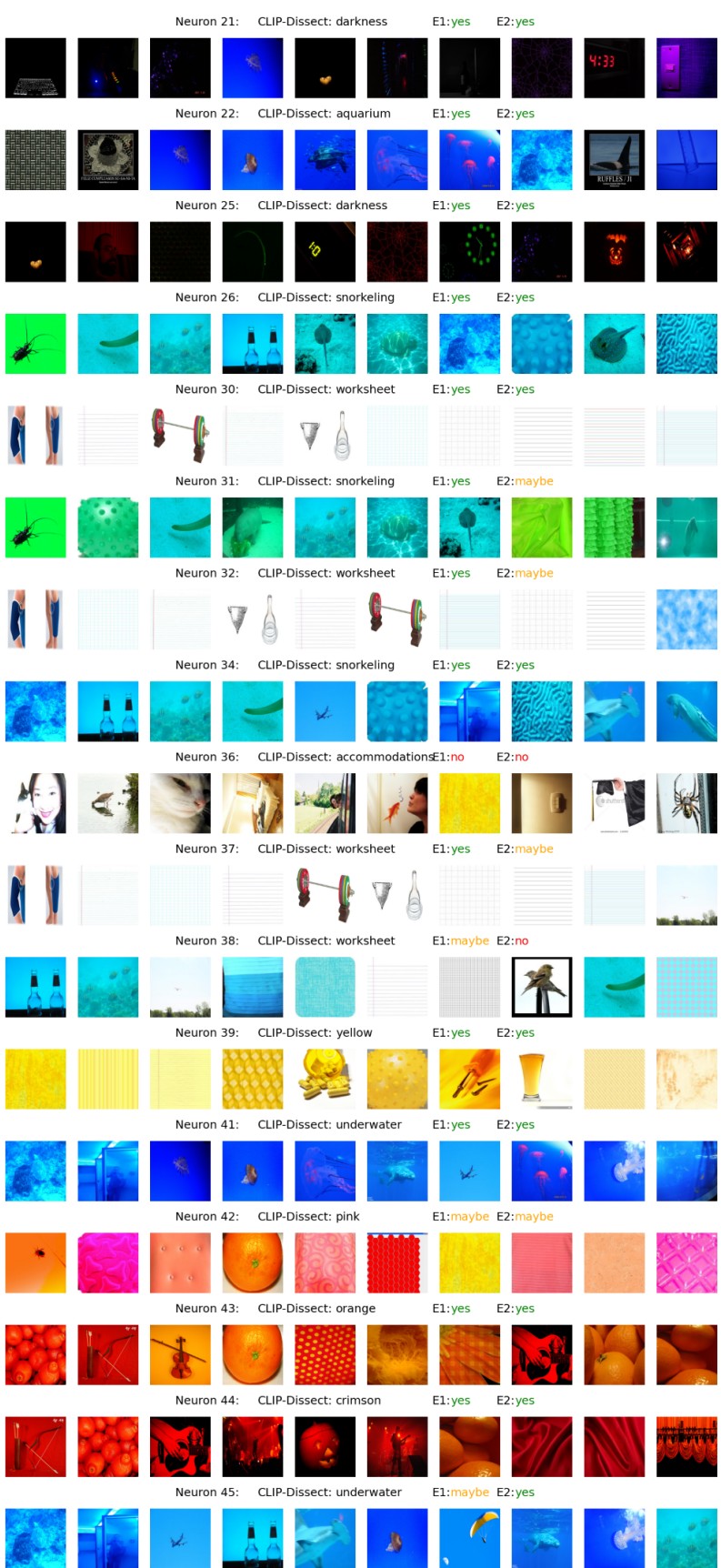

Figure 17: Random neurons of 'conv1' in ResNet-50 (ImageNet), and E1, E2's evaluation on the description of CLIP-Dissect (PART 2)

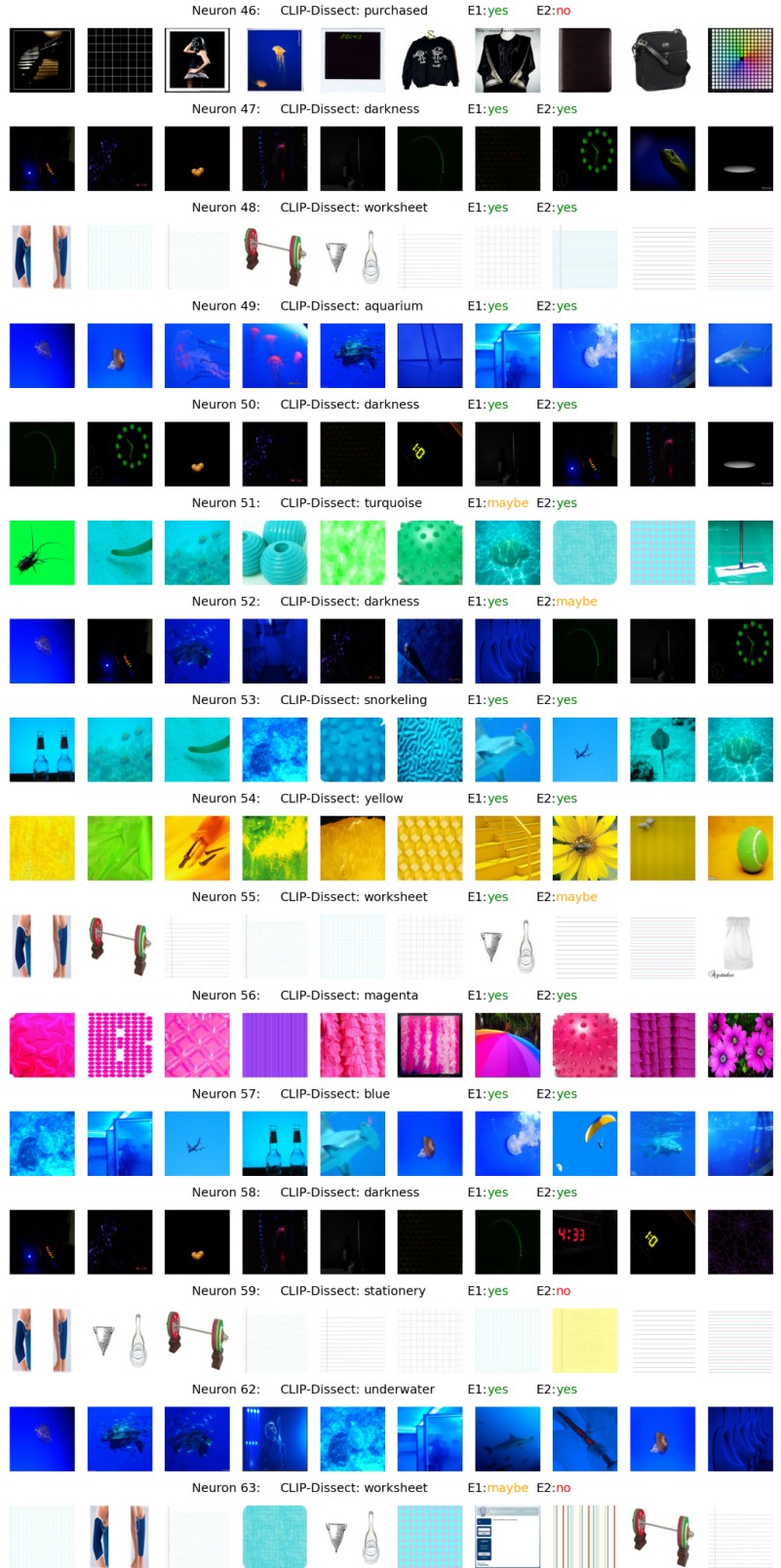

Figure 18: Random neurons of 'conv1' in ResNet-50 (ImageNet), and E1, E2's evaluation on the description of CLIP-Dissect (PART 3)

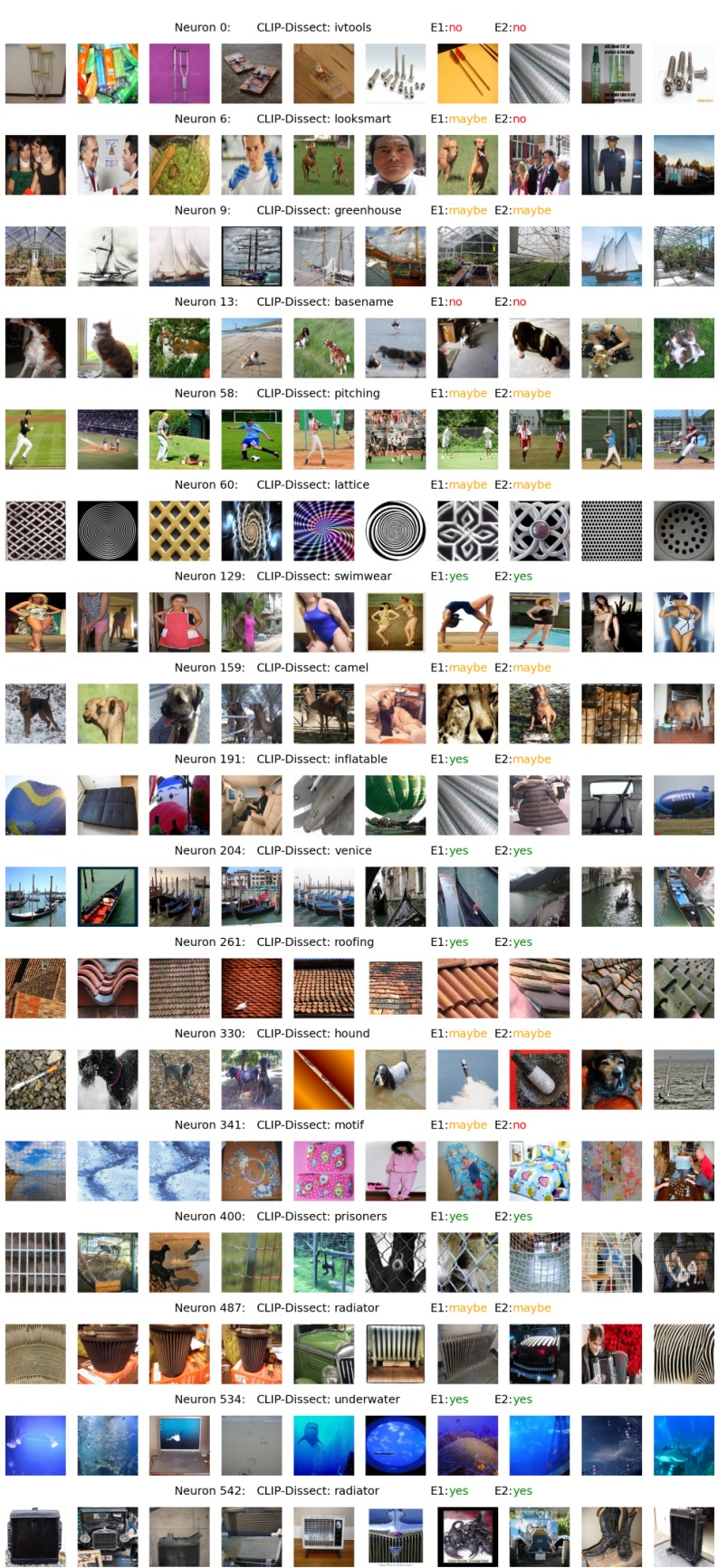

Figure 19: Random neurons of 'layer 4' in ResNet-18 (Places 365), and E1, E2's evaluation on the description of CLIP-Dissect (PART 1)

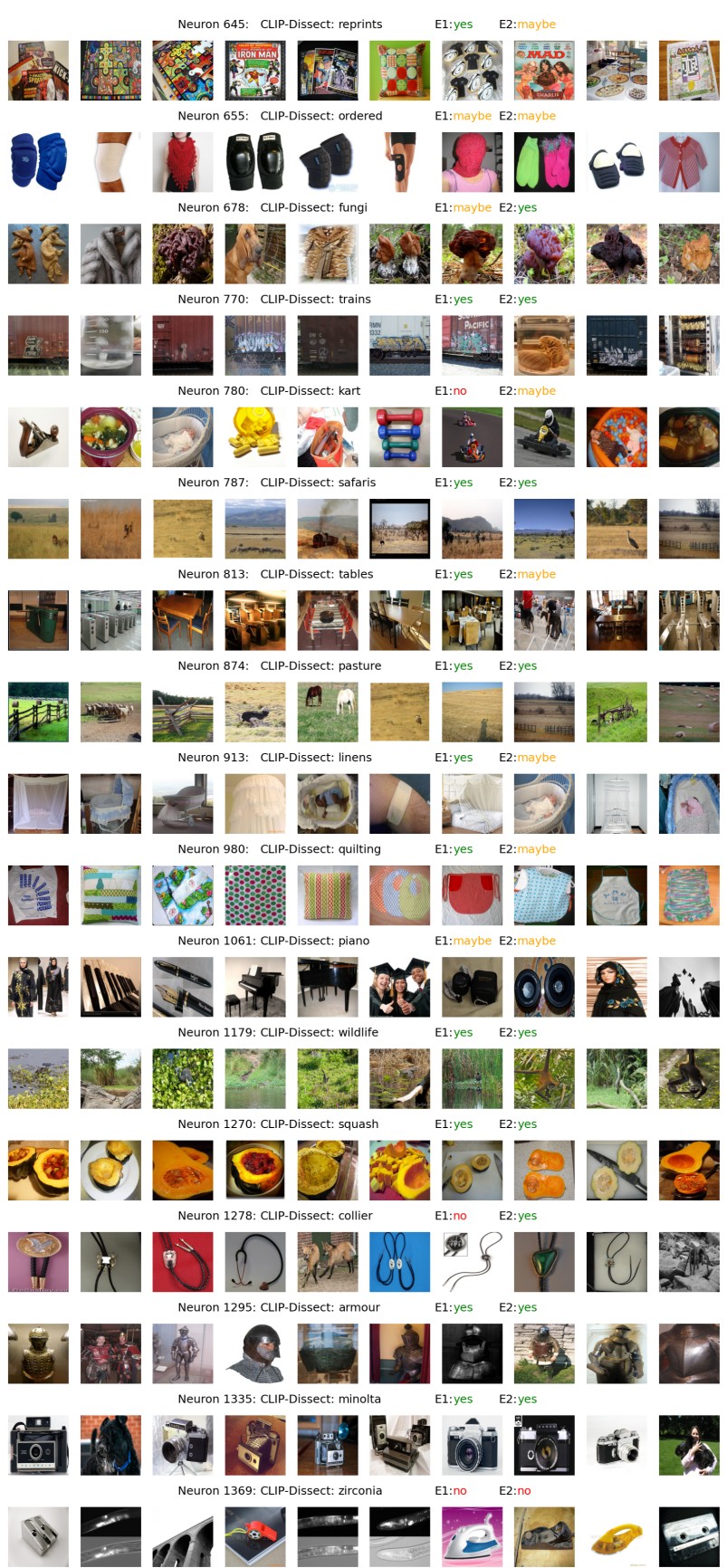

Figure 20: Random neurons of 'layer 4' in ResNet-18 (Places 365), and E1, E2's evaluation on the description of CLIP-Dissect (PART 2)

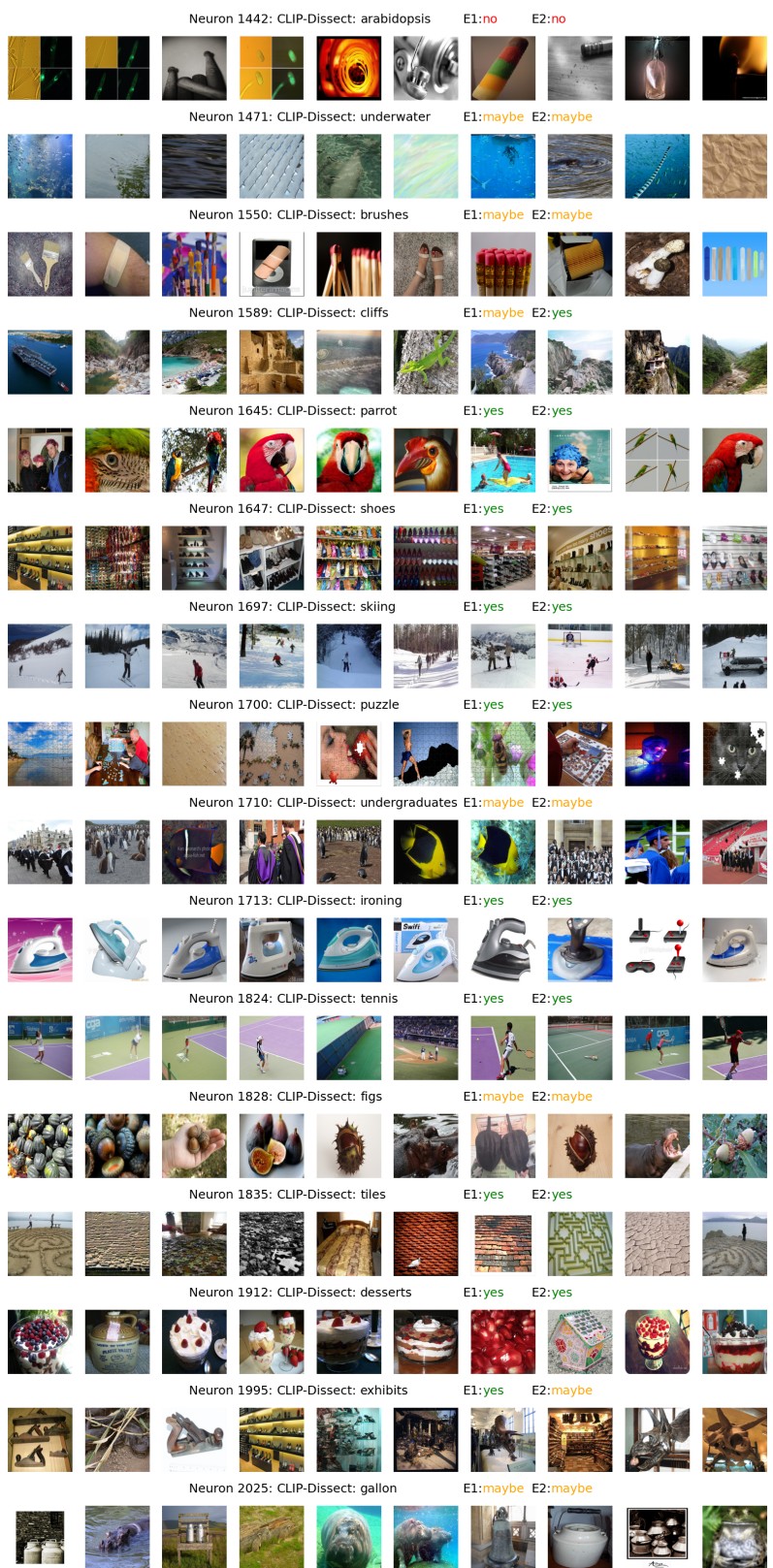

Figure 21: Random neurons of 'layer 4' in ResNet-18 (Places 365), and E1, E2's evaluation on the description of CLIP-Dissect (PART 3)

