# OpenReview forum: "CLIP-Dissect: Automatic Description of Neuron Representations in Deep Vision Networks"
_ICLR.cc/2023/Conference — ICLR 2023 notable top 25%_

### Official Review · Reviewer_Ujcx · 2022-10-21

**Confidence:** 3
**Correctness:** 3
**Technical Novelty And Significance:** 2
**Empirical Novelty And Significance:** 2
**Recommendation:** 6

**Clarity, Quality, Novelty And Reproducibility:**

Clarity: The paper is well written and easy to follow.
Reproducibility: I believe the paper is reproducible although I didn't try that.
Novelty: Novel but limited.

**Strength And Weaknesses:**

Strength
- Since the introduction of CLIP model, its powerful representation ability has been proved by a variety of applications. This paper showcases an interesting application of CLIP model.
- The experimental results are impressive and shows the effectiveness of the proposed method.

Weaknesses
- The contribution of the paper may be exaggerated. The superior results over existing methods may be largely attributed to the powerful representation ability of CLIP that is pretrained on hundreds of millions of image-text pairs.

**Summary Of The Paper:**

This paper proposes a network dissection method without using any labelled image data. This is achieved by utilizing a pretrained CLIP model to compute an image-concept similarity matrix. With the help of the matrix, the concept label for a neuron unit will be the one that maximizes the similarity between the neuron query and concept vector.

**Summary Of The Review:**

Overall, this paper is an interesting application of CLIP model, but the main reason it works well is the pretrained CLIP's powerful representation ability.

---

> ### Author Response · Authors · 2022-11-17
> **Author response**
>
> Thank you for the review! We would like to discuss your concerns below:
>
> **#1 Whether the results are mainly achieved by the capability of CLIP pretrained model**
>
> We thank you for the comments and while we acknowledge that the capabilities of CLIP are very important to our results, we think our paper has many valuable contributions by itself. The way we see this question is that we have developed a tool for describing neuron functionality that performs better than previous tools. Regardless of whether the good performance of CLIP-dissect is attributed more to CLIP itself or the novelty of our methods, we think that it is unlikely people would use CLIP this way without reading our manuscript, therefore we believe our results are important to share and will benefit the research community.
>
> In particular we think the following contributions are valuable to the research community:
>
> - We are the first to show that CLIP can be used very effectively to describe functionality of individual neurons
> - We develop a new and efficient pipeline as well as principled similarity functions (sec 3.2) that are critical in utilizing CLIP for this task, and showcase their importance
> - We propose two new automated methods (sec 4.2, Table 1-2; Appendix B.2) for evaluating the quality of neuron descriptions. These methods were critical in our ability to find a well performing similarity function for CLIP-Dissect (Table 3).
> - We provide extensive empirical evaluation with more than 10 different experiments, showcasing our ability in a large range of tasks such as: showing our method works for CNNs and Vision Transformers, a large evaluation of the description quality across different layers and discovering a link between large weights and similar functions between the connected neurons.
> - Our results also indicate that CLIP-dissect generates higher quality of neuron descriptions compared to prior work, while also being much more computationally efficient than prior work (Table 4).
>
> **Novelty of similarity function:**
> Here we discuss the effort required to make CLIP-Dissect perform well, in particular the importance of similarity function and how it is *non-trivial* to design a good one. One concrete example is Table 3 of the original manuscript, where we performed quantitative analysis of CLIP-dissect on predicting the concept of the last layer of neurons for 4 different similarity functions (cos, rank reorder, WPMI and SoftWPMI) on ImageNet. It can be seen in Table 3 that, using the naive cosine similarity (cos) with CLIP model, the accuracy is very low (8.5%-15.9% for different probing datasets), while using our theoretically-inspired similarity function SoftWPMI, the accuracy is much higher (46.2%-95.4%). The above result suggests that even if the CLIP model is very powerful, without carefully designing the similarity function the quality of the neuron concepts will be very poor. To get a better idea of the significance of this performance difference, we report additional results in Appendix B.5 Qualitative Effect of Similarity Function, where in Fig 21 we showcase the descriptions generated using cos similarity function for the same neurons that we reported in Fig 1 of the original manuscript. We can see that the descriptions generated by CLIP-dissect with cos similarity function are almost useless, with only 1/8 descriptions matching the highly activating images, while the descriptions generated using SoftWPMI are very accurate. This suggests that it is an *non-trivial* effort to design similarity functions, highlighting the need for more sophisticated similarity functions we introduced in Sec 3.2.
>
> **#2 All new experiments and changes provided in this rebuttal period**
>
> Please see our General Response: Overview of new results in a separate post for description of other new experiments performed during rebuttal.
>
> **#3 Summary**
>
> To summarize, we have:
>
> - Discussed in #1 the contributions and novelty of our method, with evidence showing that a well designed similarity functions are the key for CLIP-dissect to succeed
> - Described in #2 on all new experiments and changes in this rebuttal period
>
> We believe that we have addressed your concerns in #1 and with additional new experiments (see General response), please let us know if you still have any reservations and we would be happy to address them.

---

> > ### Comment · Reviewer_Ujcx · 2022-11-17
> > **Thanks for clarification**
> >
> > Thanks for the clarification. After reading the authors' response, I would like to change the score from 5 to 6.

---

> > > ### Author Response · Authors · 2022-11-17
> > > **Thank you!**
> > >
> > > Dear Reviewer,
> > >
> > > Thank you for the prompt reply and increasing the score, we are happy to see our response has addressed your concerns!

---

### Official Review · Reviewer_xZV4 · 2022-10-24

**Confidence:** 3
**Correctness:** 4
**Technical Novelty And Significance:** 2
**Empirical Novelty And Significance:** 3
**Recommendation:** 6

**Clarity, Quality, Novelty And Reproducibility:**

- Clarity: The paper is very clearly written and easy to follow.
- Quality: The quality of the presentation is good and the method is technically sound.
- Novelty: The paper has marginal technical novelty as it owes main capability of describing the activation to CLIP, while it has slightly more empirical novelty since this is the first attempt to utilize large vision-language model for explaining the behaviors of neurons without limiting the vocabulary to a predefined set.
- Reproducibility: I believe the sufficient information for reproducing the result is provided.


**Strength And Weaknesses:**

## Strength
1. The paper provides sufficient background so that the position of the paper is clear. The difference with relevant works such as Network Dissection is well explained. Obtaining the capability of describing role of neurons without limiting the vocabulary to a predefined set is a good contribution to the community.
1. The proposed approach is reasonable and technically sound. The method is so simple that readers can easily integrate the method into further studies in this domain.
1. The paper is clearly written and easy to follow.

## Weakness
1. The technical novelty of the proposed method is limited since the method is rather straight-forward adaptation of CLIP for network dissection problems. The paper claims that it addresses the limitations stated in the last line of the 2nd paragraph in the introduction “Although these… new concept.”, but it is achieved mainly by the capability of CLIP pretrained model. In addition, the results presented in the experiment section is somewhat “as expected” given the CLIP’s capability. I acknowledge the paper’s empirical contribution, but the contribution in terms of the knowledge advancement in the community might be a little lower compared to standard ICLR papers. I would appreciate if the authors could discuss the novelty of the paper and new findings brought by the paper to the community.

Minor points.
1. The name “CLIP-Dissect” is emphasized with the bold font everywhere it appears, which I think is unnecessarily distracting.
1. p3 “STOA” -> “SOTA”
1. Please provide the citation for WPMI.
1. p5 “among the three” -> “among the four”
1. Section 4.3: “SoftPMI” -> “SoftWMPI”


**Summary Of The Paper:**

This paper proposes a new method for describing the representations obtained in each neuron of neural networks. Unlike previous methods that require a dedicated dataset with annotations and thus can only detect concepts that appear in the dataset, the proposed method CLIP Dissect is able to describe the representation of neurons on the basis of the implicit knowledge stored in the pretrained large-scale vision-language model (CLIP). The experiment reveals that the proposed method can not only achieve the aforementioned goal, but also provide better descriptions than previous methods even on the predefined datasets.

**Summary Of The Review:**

In my view, although the technical contribution of the paper is not that significant, the paper provides important extension in understanding the representations of neurons in neural networks. In addition, the paper clearly states its position and writing quality is good. Overall, I am leaning to positive on this paper, but I am open to the discussion with the authors and the other reviewers to make the value of this paper clearer.

---

> ### Author Response · Authors · 2022-11-17
> **Author response**
>
> Thank you for the comments and thoughtful review! Please see our response to your question below.
>
> **#1 Whether the results are mainly achieved by the capability of CLIP pretrained model**
>
> We thank you for the comments and while we acknowledge that the capabilities of CLIP are very important to our results, we think our paper has many valuable contributions by itself. The way we see this question is that we have developed a tool for describing neuron functionality that performs better than previous tools. Regardless of whether the good performance of CLIP-dissect is attributed more to CLIP itself or the novelty of our methods, we think that it is unlikely people would use CLIP this way without reading our manuscript, therefore we believe our results are important to share and will benefit the research community.
>
> In particular we think the following contributions are valuable to the research community:
>
> - We are the first to show that CLIP can be used very effectively to describe functionality of individual neurons
> - We develop a new and efficient pipeline as well as principled similarity functions (sec 3.2) that are critical in utilizing CLIP for this task, and showcase their importance
> - We propose two new automated methods (sec 4.2, Table 1-2; Appendix B.2) for evaluating the quality of neuron descriptions. These methods were critical in our ability to find a well performing similarity function for CLIP-Dissect (Table 3).
> - We provide extensive empirical evaluation with more than 10 different experiments, showcasing our ability in a large range of tasks such as: showing our method works for CNNs and Vision Transformers, a large evaluation of the description quality across different layers and discovering a link between large weights and similar functions between the connected neurons.
> - Our results also indicate that CLIP-dissect generates higher quality of neuron descriptions compared to prior work, while also being much more computationally efficient than prior work (Table 4).
>
> **Novelty of similarity function:**
> Here we discuss the effort required to make CLIP-Dissect perform well, in particular the importance of similarity function and how it is *non-trivial* to design a good one. One concrete example is Table 3 of the original manuscript, where we performed quantitative analysis of CLIP-dissect on predicting the concept of the last layer of neurons for 4 different similarity functions (cos, rank reorder, WPMI and SoftWPMI) on ImageNet. It can be seen in Table 3 that, using the naive cosine similarity (cos) with CLIP model, the accuracy is very low (8.5%-15.9% for different probing datasets), while using our theoretically-inspired similarity function SoftWPMI, the accuracy is much higher (46.2%-95.4%). The above result suggests that even if the CLIP model is very powerful, without carefully designing the similarity function the quality of the neuron concepts will be very poor. To get a better idea of the significance of this performance difference, we report additional results in Appendix B.5 Qualitative Effect of Similarity Function, where in Fig 21 we showcase the descriptions generated using cos similarity function for the same neurons that we reported in Fig 1 of the original manuscript. We can see that the descriptions generated by CLIP-dissect with cos similarity function are almost useless, with only 1/8 descriptions matching the highly activating images, while the descriptions generated using SoftWPMI are very accurate. This suggests that it is an *non-trivial* effort to design similarity functions, highlighting the need for more sophisticated similarity functions we introduced in Sec 3.2.
>
> **#2 minor points**
>
> Thank you for the suggestions. In the updated submission we have:
> - removed bolding from references to CLIP-Dissect
> - Added citation to WPMI
> - Fixed the 3 typos
>
> **#3 All new experiments and changes provided in this rebuttal period**
>
> Please see our General Response: Overview of new results in a separate post for description of other new experiments performed during rebuttal.
>
> **#4 Summary**
>
> To summarize, we have:
> - Discussed in #1 on the contributions and novelty of our method, with evidence showing that a well designed similarity functions are the key for CLIP-dissect to succeed
> - Corrected in #2 on the typos and formatting
> - Described in #3 on all new experiments and changes in this rebuttal period
> Please let us know if you have any additional questions or comments, we would be happy to discuss further.

---

> > ### Comment · Reviewer_xZV4 · 2022-11-18
> > **Thank you for the feedbac,**
> >
> > I appreciate the authors' feedback.
> > After reading the response as well as the other reviewers' comments, I am still positive about the paper.

---

> > > ### Author Response · Authors · 2022-11-18
> > > **Thank you**
> > >
> > > Thank you for the positive comments and quick response!

---

### Official Review · Reviewer_XsCd · 2022-10-24

**Confidence:** 4
**Correctness:** 3
**Technical Novelty And Significance:** 3
**Empirical Novelty And Significance:** Not applicable
**Recommendation:** 8

**Clarity, Quality, Novelty And Reproducibility:**

**Clarity:** I found the work to be written well and clearly.

**Quality:** The experiments are well-designed, and the paper is also well-presented.

**Reproducibility:** Their approach is well described. However, the authors did not comment on whether they plan to release their source code.

**Novelty:** As far as I know, CLIP embeddings were not previously used to label internal activations. However, the idea is also not revolutionary.

**Minor issues:** The tables' captions should be above the table.

**Strength And Weaknesses:**

**Strengths:**

- The paper cleverly uses CLIP embeddings to generate labeled annotations for neurons.
- Labeling internal neurons could be very useful for many downstream applications. For example, it could be used to monitor networks in production (e.g., does new data change the internal activations in unintended ways?).
- The evaluation contains a good collection of qualitative and quantitative experiments.
- Even though this new approach does not require any labels, it outperforms previous label-based approaches.

**Weakness:**

- My main concern is that this work did not measure the gap between the assigned meaning (as textual description) and actual meaning (as neuron activations). Previous works in that direction also did not analyze this gap in detail. For example, the following experiments could be used to measure this gap:

  - How well can the final classes be predicted using the textual description of the internal neurons? This could also be used to evaluate different labeling strategies: the better the description, the higher the final score should be.
  - Show that the most activating neurons correspond to the assigned concepts. Just providing the 5-most activating examples is not enough. I would like to see a uniform selection of most-activating samples from the top 0.1%,1%, and 5%-most activating examples per neuron. I would suspect that already the top 1% are quite diverse.

- Similar to the previous comment: In Appendix A.3, it is stated: *"Unintrepretable neurons: Some neurons seem to not be interpretable"*. Could you provide a qualitative estimate on how many neurons are "uninterpretable" and how you measure this?

- Limitations are only discussed in Appendix A.2. Please move them to the main paper. There is enough whitespace on page 9 to include them there. Furthermore, it should be discussed how well the meaning of a neuron matches the textual description.

##

**Summary Of The Paper:**

The paper proposes a new way to label internal neurons of convolutional neural networks. The proposed method relies on the CLIP's vision and text embedding to generate the labels. This is an improvement over related work that relied on manually labeled datasets. For the evaluation, two networks are used: a ResNet-50 trained on ImageNet and a ResNet-18 for the Places-365 dataset. Their method is compared against two baseline methods on different datasets and tasks.

**Summary Of The Review:**

I think this paper makes interesting contributions and could be a good fit for ICLR. However, I doubt a few words can fully describe a neuron's functionality, and I think this mismatch must be analyzed or at least discussed.

---

> ### Author Response · Authors · 2022-11-17
> **Author Response 1/2**
>
> Thank you for the positive feedback and suggestions to help us improve the work! We would like to address your remaining concerns to the best of our ability with below new experiments. Please see below new results and discussion.
>
> **#1 Did not measure the gap between the assigned meaning and actual meaning**
>
> We would like to clarify that we think we do address this issue unlike previous work. In particular, our experiments in section 4.2 (Tables 1 and 2) measure specifically this. In the section we apply our method to describe the final layer neurons of a network, where we know the ‘actual meaning’ of those neurons, which is to detect their assigned class. We can then compare how well the text descriptions we have created match this ‘actual meaning’.
>
> **#2 Predicting final classes using internal neuron descriptions**
>
> Following your suggestion, we have conducted an experiment trying to use neuron descriptions of the neurons with the highest contribution to the decision to predict the final class label. In particular we studied whether the description of the neuron with the highest contribution to current decision matches the ground truth class of that image, by predicting the class with the highest similarity to the description of the most important neuron.
>
> In our experiment with ResNet-50 (ImageNet) we found that using CLIP-Dissect descriptions we were able to correctly predict the class of 10.28% of the images. In contrast, when we used descriptions from Network Dissection we were only able to predict the class of 3.36% of the images, and with MILAN(base) only 2.31% of the time. It’s worth noting that overall we would not expect to reach a very high predictive accuracy using the above method as the most contributing neuron often does have a completely different role than the target class. However if we study the same network with different neuron description methods, we would expect a better neuron description method would return a higher prediction accuracy, assuming at least some of the important neurons indeed do have similar role to the class itself. This method has the benefit of being automated, going beyond visualizing few most highly activating images and being able to analyze hidden layer description quality. We think methods like this are an interesting future direction for evaluating neuron description methods. Full experimental details and results are discussed in Appendix B.2.
>
> **#3 Uniformly selecting most-activating samples from 0.1%, 1% and 5% examples**
>
> Following your suggestion, we have conducted the experiment and the results are reported in Appendix B.3 Visualizing Wider Range of Activations. We visualize the most interpretable neurons of two layers of ResNet-18 (Places 365) and ResNet-50 (ImageNet) by uniformly sampling images from the top 0.1%, 1% 5% of most highly activating images in Figures 19 and 20.
>
> We can see that as you thought the descriptions tend to match quite well for the 0.1% of most highly activating images, but the top 1% and top 5% images start to be of quite different concepts only slightly related to the description. We also notice that low level concepts like colors tend to be more consistently represented in top 1% and 5% images while higher level concepts are not. This result is somewhat expected as the $\mathcal{D}_{\textrm{probe}}$ does not have that many images for each higher level concept, but highlights the need to explore neuron activations beyond just the few most highly activated images.
>
> **#4 Measuring (un)interpretable neurons**
>
> We follow a similar definition procedure to Network Dissection to define a neuron as interpretable when the similarity value is greater than a predefined threshold $\tau$. (In Network Dissection they used IoU>0.04). Accordingly, when the similarity value is less than this threshold $\tau$, then a neuron is said to be uninterpretable.
>
> In Appendix B.4 Interpretability Cutoff, we discuss this cutoff in more detail, and report the cut-off threshold $\tau$ for our best performing similarity function SoftWPMI proposed in Sec 3.2, which is 0.16. This cut-off is selected based on the description score of the neurons from human evaluators (please see Appendix B.1 for experiment details). With this cut-off, we find that find 69.7% of neurons in ResNet-18 (Places-365) and 77.8% of neurons in ResNet-50 (ImageNet) to be interpretable, indicating that around 22-30% of neurons do not have a simple explanation for their functionality, i.e. are 'uninterpretable'.

---

> > ### Author Response · Authors · 2022-11-17
> > **Author response 2/2**
> >
> > **#5 Limitations location**
> >
> > Unfortunately we were unable to fit limitations into the main text without violating the page limit, but we have moved them up to Appendix A.1 for increased visibility.
> >
> > **#6 whether using a few words can fully describe a neuron’s functionability**
> >
> > Thank you for your comment and insight! We agree with you that indeed it could be the case that some neurons have multiple functionality and require multiple concepts or natural language sentences to describe the neuron’s functionality more accurately. In fact, our proposed approach can be easily extended to multiple concepts with logical expression (e.g. Leaf AND Cloud; Water NOT Red, etc) and natural language description (e.g. flying birds in the sky), by modifying the concept set S or composing multiple explanations as discussed in section 4.6, but this might still not be enough to capture more subtle functions of neurons. We have added discussion of this to Limitations in section A.1 (marked in blue).
> >
> > **#7 Releasing source code**
> >
> > We would like to clarify that we mentioned that “we will make all code publicly available prior to publication” under Sec 7 Reproducibility in the original draft.
> >
> > **#8 Table captions**
> >
> > Thank you for pointing this out. We have moved captions above the table in the revised submission.
> >
> > **#9 All new experiments and changes provided in this rebuttal period**
> >
> > Please see our General Response: Overview of new results in a separate post for description of other new experiments performed during rebuttal.
> >
> > **#10 Summary**
> >
> > In summary we have:
> > - Clarified in #1 on how our proposed method of evaluating descriptions of final layer neurons addresses the gap between neuron label and actual function
> > - Provided in #2 a new experiment predicting the class of inputs based on descriptions of highly activating neurons, and showcased CLIP-Dissect is much more accurate than MILAN or Network Dissection
> > - Provided in #3 new experimental results visualizing wider range of highly activating images for neurons
> > - Discussed in #4 our ability to measure which neurons are uninterpretable, and found 20-30% of neurons to be uninterpretable
> > - Moved in #5 Limitations discussion up to A.1
> > - Discussed in #6 the limitations of few word descriptions
> > - Clarified in #7 that we will release source code
> > - Moved in #8 table captions above the table
> > - Described in #9 on all new experiments and changes in this rebuttal period
> >
> > Please let us know if you have any additional questions or comments, we would be happy to discuss further.

---

> > > ### Comment · Reviewer_XsCd · 2022-11-18
> > > **Re: Rebuttal**
> > >
> > > I want thank the authors to conduct the additional experiments. This shows a gap between assigned description and actual meaning, but I see this rather as a strength of this work as it could be the first work to discuss this gap in this research area.
> > >
> > > The accuracy of 10.28% in the prediction experiment (Appendix B.2) also shows that there is a decent amount of improvement for future work. However, this might only be resolved if additional constraints are applied during the training of the explained models.
> > >
> > > I would still insist to mention this gap in the main text. As argued before, I think this could make your work stronger. If you have space restrictions, you could move Table 4 to the appendix (it might not even be necessary at all as the numbers are also in the  text).
> > >
> > > ----------
> > > I also read the other reviews and rebuttals and I am still convinced that this work should be accepted to ICLR. I will also increase my score to reward the additional experiments and scientific honesty to report the gap between assigned description and actual meaning.

---

> > > > ### Author Response · Authors · 2022-11-18
> > > > **Thank you**
> > > >
> > > > Thank you for the quick response and positive feedback! Following your suggestion we have uploaded a new version of the manuscript where we moved Limitations into the main text (section 6, page 9) and Table 4 into the Appendix.

---

### Official Review · Reviewer_SGoY · 2022-10-26

**Confidence:** 5
**Correctness:** 3
**Technical Novelty And Significance:** 2
**Empirical Novelty And Significance:** 3
**Recommendation:** 8

**Clarity, Quality, Novelty And Reproducibility:**

The method is well presented and the paper is well written and easy to follow.  The method is easy to reproduce and very original in its use of strong pre-trained vision and language models to a completely new research area.

**Strength And Weaknesses:**

Strengths:
1. The paper demonstrate a new and non trivial use case of strong pre-trained Vision and Language models which is, in my opinion, a contribution by itself to the field of joint multi-modal embedding.

2. The proposed method is far more flexible than previous methods - leveraging CLIP's latent space, the method can work on practically any set of natural images and concepts without requiring any training or ground truth labels. The method is also dramatically faster than previous methods.

3. The experiments demonstrate the methods flexibility and improved performance (compared to existing methods). The paper also demonstrates two non-trivial use cases of the method: detecting missing concepts from a given image set and investigating the connection between pairs different neurons in a pre-trained network.


Weaknesses:
EDIT: the rebuttal and the revised text (additional experiments and analysis) completely addressed the drawbacks I mentioned below.

1. In my opinion, the quantitative analysis basically measures how well the method assigns the correct class labels for the neurons in the final layer, which is a very limited aspect of explainability. The authors discuss the limitations of different evaluation methods in the paper, but the fact is that there is no quantitative experiment evaluating the ability of the network to describe intermediate neurons in the network. I understand the limitations of having no ground truth, but I do believe designing a small user study could benefit the paper.

2. minor comment: under "Rank reorder" similarity function, it says "This is done by replacing the i-th largest element of P:,m by the i-th largest element of qk for all i.". The resulting vector is a reordered version of q_k, wouldn't that mean that we replace the i'th largest element of q_k by the i'th largest element of P:,m? If I understand correctly, "Rank reorder means re-ordering q_k using P:,m order", I think the phrasing of this section can be made a bit more clear.


**Summary Of The Paper:**

The paper addresses deep neural network explainability and specifically proposes a method for generating textual description of neurons in a pre-trained network. The method operates as follows: Given a pretrained network f, a "probe set" D (a set of images), a "concept set" S (a set of words/phrases) and a neuron in f, the method finds the concept in S that best describe the neuron, in terms of explaining what it "does"/responds to" based on its activation map on the probe set D. The method works as follows:

1. First, we use CLIP to compute image embeddings for D and text embeddings for S.
2. Next, we compute the concept activation matrix P where the i,j entry in P is the the result of the inner product of the i'th image embedding with the j'th concept embedding.
3.  For a given neuron k, we compute the mean of its activation on map on all the images in D and denote the resulting vector as q_k.
4. Finally, we find the "most similar" concept t_l (in S), where the similarity function is computed using q_k and the matrix P. The paper explores several alternatives for a similarity function and provides detailed experiments comparing the different alternatives.

The paper provides quantitative as well as qualitative evaluations that demonstrate the improve performance, flexibility and runtime of the proposed method.

**Summary Of The Review:**

In my opinion, the paper presents a useful, general, flexible method fast method for addressing an important aspect of model explainability. The technical novelty of the paper is limited in the sense that it proposes a simple method which relies heavily on the strength of CLIP, but I do think the paper is good fit for ICLR.

---

> ### Author Response · Authors · 2022-11-17
> **Author response**
>
> Thank you for the positive feedback and thoughtful review to help us improve the work! Please see below our response to your comments.
>
> **#1 quantitative analysis for hidden neurons**
>
> Following your suggestion, we have conducted a more quantitative experiment on the quality of descriptions on hidden layer neurons and the results are reported in *Appendix B.1 Larger scale experiment on description quality*. We have evaluated the description quality of 50 randomly selected neurons for each of the 5 layers and 2 models studied, for a total of 1000 evaluations. Each evaluator was presented with 10 most highly activating images, and answered the question: "Does the description: { } match this set of images?". An example of the user interface is shown in Figure 12. Each evaluation had three options which we used with the following guidelines:
>
> Yes - Most of the 10 images are well described by this description
>
> Maybe - Around half (i.e. 3-6) of the images are well described, or most images are described relatively well (accurate but too generic, or slightly inaccurate)
>
> No - Most images are poorly described by this caption
>
> These evaluations were turned into a numeric score with the following formula: yes:1, maybe:0.5, no:0. Table 5 shows the average description score across different neurons and evaluators for each of the layers evaluated. This average score can be thought of as the percentage of neurons well described.
>
> We observed that overall the descriptions are good for 55-80% of neurons depending on the layer, with the average score across all evaluations being 0.655. In addition we notice that the very early and very late layers are most interpretable, corresponding to clear low or high level concepts, while the middle layers seem to be harder to describe. It is worth noting that we are evaluating random neurons here, i.e. the neurons are selected randomly, so the displayed neuron may not be interpretable in the first place -- in many cases when the description does not match are because the neuron itself is not 'interpretable', i.e. there is no simple description that corresponds to the neurons functionality.
>
> It can be seen that even with the evaluation guidelines described above, these evaluations are somewhat subjective; nevertheless, we found that our evaluators agreed on 68.4% of the neurons with the vast majority of disagreements being between yes/maybe or maybe/no, with only 2.4% of neurons having a yes from one evaluator and no from another. For transparency, we have included all 50 neurons, their descriptions and two evaluator's evaluations (E1, E2) of these descriptions for ResNet-50 layer conv1 in Figures 13-15 and for ResNet-50 layer4 in Figures 16-18. In addition we have included the full results for all 10 layers in the supplementary material for full transparency.
>
> **#2 Clarification on rank reorder similarity function**
>
> Thanks for the comment! The idea of rank reorder is to first generate a new vector $q_k'$, which is a re-ordered version of the vector $q_k$ according to the rank of the elements in the vector $P_{:,m}$. Then, in the similarity function, we compute the negative norm distance between $q_k'$ and $q_k$ as mentioned in Eq (2), which gives high similarity when $q_k$ and $P_{:,m}$ have similar order of values. It can be thought of to first copy the vector $q_k$ to the new vector $q_k'$, and then reorder the elements of $q_k'$ according to the order in the vector $P_{:,m}$. We have clarified this in the updated submission in Section 3.2 (page 4), with new text marked in blue.
>
> **#3 All new experiments and changes provided in this rebuttal period**
>
> Please see our General Response: Overview of new results in a separate post for description of other new experiments performed during rebuttal.
>
> **#4 Summary**
>
> To summarize, we have:
> - Provided in #1 a large scale analysis on the quality of our neuron descriptions, with 1000 evaluations of description quality and have included full results.
> - Clarified in #2 the description for rank reorder similarity function
> - Described in #3 on all new experiments and changes in this rebuttal period
>
> Please let us know if you have any additional questions or comments, we would be happy to discuss further.

---

> > ### Comment · Reviewer_SGoY · 2022-11-22
> > **Response to rebuttal**
> >
> > I would like to thank the authors for the detailed rebuttal, the added experiments and the changes made to the paper. In my opinion, the additional experiments presented throughout appendix B make the paper much stronger (and I'm referring to all the experiments, not only B.1). I have therefore updated the score given to 8 - strong accept.

---

> > > ### Author Response · Authors · 2022-11-22
> > > **Thank you!**
> > >
> > > Thank you for the response, we are happy to hear you find our submission stronger! We would like to thank you and the other reviewers for all the productive feedback that helped us improve the paper.

---

### Author Response · Authors · 2022-11-17
**General Response: Overview of new results**

In response to reviewer comments, we have conducted several new experiments and made some changes to existing manuscript content, this response is a summary of these. New text is written in blue for visibility in the PDF.

## New experiments
We have compiled all the new results and discussion in the Appendix B (p.20-p.31) in the PDF with a below short summary:
- In Appendix B1, we do a larger study on the quality of descriptions on hidden layer neurons based on human evaluations, evaluating the descriptions for 500 random neurons from 10 different layers of 2 models. We observed that overall the CLIP-dissect descriptions match human evaluation for 55-80% of neurons in different layers. We have included the neurons and evaluation results for 2 layers in Figures 13-18, and the rest in the supplementary material.
- In Appendix B2, we additionally study whether the highly activating neurons in internal layers can be used to predict what class the input is from. We observed when we used descriptions from CLIP-dissect, the prediction accuracy is higher than the descriptions from other baseline methods (Network dissection, MILAN), suggesting our method gives higher description quality than the baselines.
- In Appendix B3, we visualize the most interpretable neurons of two layers of ResNet-18 (Places 365) and ResNet-50 (ImageNet) by uniformly sampling images from the top 0.1%, 1% 5% of most highly activating images in Figs 19 and 20. This shows our descriptions mostly describe top 0.1% images but beyond that highly activating images start to diverge.
- In Appendix B4, we discuss using our similarity score to evaluate whether neurons are interpretable, and using a cutoff we found observed that 69.7% of neurons in ResNet-18 (Places-365) and 77.8% of neurons in ResNet-50 (ImageNet) to be interpretable.
- In Appendix B5, we additionally compare CLIP-dissect with naive cos similarity v.s. proposed SoftWPMI, for the purpose of demonstrating that CLIP-dissect may work poorly if the similarity function is not designed well, highlighting the need for more sophisticated similarity functions we introduced in Sec 3.2

## Other Changes
- Clarified similarity function descriptions in Section 3.2
- **Edit:** Moved discussion on Limitations into main text for more visibility while moving Table 4 into the Appendix
- Added discussion on limits of short neuron descriptions to Limitations
- Moved table captions above the tables
- Fixed typos

---

### Author Response · Authors · 2023-06-05
**Update: Crowdsourced evaluation results**

We have conducted an additional crowdsourced evaluation of the neuron description quality. The results are available in the Appendix B on arxiv: https://arxiv.org/abs/2204.10965

---

### Decision · Program_Chairs · 2023-01-20

**Decision:**

Accept: notable-top-25%

**Justification For Why Not Higher Score:**

The task is a bit niche, and there is not much to learn from the proposed method technically (which is fine).

**Justification For Why Not Lower Score:**

I does stand out a bit from the typical work and I think deserves the spotlight.

**Metareview: Summary, Strengths And Weaknesses:**

The reviewers were all (after the rebuttal and discussion) positive, and in particular appreciated the novel use of the language/vision models this paper proposes, and the well designed evaluation (both in the submission and in subsequent exchanges). I think this is very interesting work that may open up an avenue for subsequent efforts in the area of automated discovery and explainability.



**Note From Pc:**

if the above contains the word "oral" or "spotlight" please see: "oral" presentation means -> notable-top-5% and "spotlight" means -> notable-top-25%. As stated in our emails, we are disassociating presentation type from AC recommendations